# A non-canonical mechanism for Crm1-export cargo complex assembly

Ute Fischer[1], Nico Schäuble[1], Sabina Schütz[1,2], Martin Altvater[1,2], Yiming Chang[1], Marius Boulos Faza[1], Vikram Govind Panse[1]*

[1]Institute of Biochemistry, Department of Biology, ETH Zurich, Zurich, Switzerland; [2]Molecular Life Science, Graduate School, Zurich, Switzerland

**Abstract** The transport receptor Crm1 mediates the export of diverse cargos containing leucine-rich nuclear export signals (NESs) through complex formation with RanGTP. To ensure efficient cargo release in the cytoplasm, NESs have evolved to display low affinity for Crm1. However, mechanisms that overcome low affinity to assemble Crm1-export complexes in the nucleus remain poorly understood. In this study, we reveal a new type of RanGTP-binding protein, Slx9, which facilitates Crm1 recruitment to the 40S pre-ribosome-associated NES-containing adaptor Rio2. In vitro, Slx9 binds Rio2 and RanGTP, forming a complex. This complex directly loads Crm1, unveiling a non-canonical stepwise mechanism to assemble a Crm1-export complex. A mutation in Slx9 that impairs Crm1-export complex assembly inhibits 40S pre-ribosome export. Thus, Slx9 functions as a scaffold to optimally present RanGTP and the NES to Crm1, therefore, triggering 40S pre-ribosome export. This mechanism could represent one solution to the paradox of weak binding events underlying rapid Crm1-mediated export.

## Introduction

In all eukaryotes, transport between the nucleus and the cytoplasm is channeled through nuclear pore complexes (NPCs) embedded within the nuclear envelope (*Tran and Wente, 2006*; *D'Angelo and Hetzer, 2008*). Yeast NPCs are approximately 60 MDa (*Fernandez-Martinez et al., 2012*) and are composed of multiple copies of about 30 nucleoporins (*Rout et al., 2000*). Cargos of different sizes and charges pass through the central transport channel of the NPC, which is filled with a meshwork of natively unfolded Phe-Gly (FG)-repeats present in FG-nucleoporins (*Frey and Görlich, 2007*; *Terry and Wente, 2009*). These nucleoporins generate a permeability barrier that allows passive diffusion of small molecules, such as ions and metabolites (*Frey and Görlich, 2007*; *Patel et al., 2007*). Macromolecules (>40 kDa) require the assistance of nuclear transport receptors, including members of the importin-β-like family, to efficiently overcome this selectivity barrier (*Macara, 2001*; *Ribbeck and Görlich, 2001*; *Rout and Aitchison, 2001*). These transport receptors, also termed importins and exportins, mediate the majority of molecular exchange between the nucleus and cytoplasm (*Cook et al., 2007*). Transport receptors recognize their cargo via specific signal sequences (*Cook and Conti, 2010*; *Xu et al., 2010*; *Güttler and Görlich, 2011*) and translocate them through the NPC by transiently interacting with FG-repeats.

The small GTPase Ran coordinates the movement of importins and exportins between the nucleus and the cytoplasm, and directs the compartment-specific binding and release of transported cargos (*Fried and Kutay, 2003*; *Pemberton and Paschal, 2005*; *Cook et al., 2007*). Ran exists in both GDP- and GTP-bound forms, and the two states are asymmetrically distributed, with RanGTP significantly enriched in the nucleus (*Nakielny and Dreyfuss, 1999*; reviewed in *Görlich and Kutay, 1999*). This gradient of RanGTP is established through the spatial separation of regulators of the Ran-cycle (*Izaurralde et al., 1997*). Whereas the Ran guanine nucleotide exchange factor, RCC1 (Prp20 in yeast)

*For correspondence: vikram.panse@bc.biol.ethz.ch

**Competing interests:** The authors declare that no competing interests exist.

**eLife digest** Plants, fungi, and animals store their genetic material within the nucleus of each of their cells. This structure is surrounded by a double layer of membrane that prevents the contents of the nucleus from mixing with the contents of the rest of the cell (namely the cytoplasm). Exchange of material between the nucleus and cytoplasm occurs through pores embedded within the nuclear membrane.

To travel through one of these pores, large molecules (also called cargos) require the assistance of so-called 'transport receptors' such as the Crm1 protein. This protein recognizes and binds to the part of a cargo molecule called a 'nuclear export signal', and the Crm1 protein also binds to another protein called RanGTP. Nuclear export signals bind weakly to Crm1, which in turn ensures that these cargos are easily released in the cytoplasm once transport is completed. However, this weak binding means that it has remained a mystery how Crm1 is able to efficiently transport cargos out of the nucleus to begin with.

Now, Fischer et al. have analyzed how one cargo that contains a nuclear export signal, namely molecules called ribosome precursors, assembles with Crm1. The experiments identified another protein called Slx9 that shuttles rapidly between the inside and the outside of the nucleus. Fischer et al. observed that Slx9 binds directly to RanGTP and brings it together with the ribosome precursor cargo. When a complex of Slx9, RanGTP, and cargo is assembled, it further recruits Crm1 to the cargo. Thus, Slx9 acts as a scaffold to bring cargo into contact with Crm1 and RanGTP, and a tight complex is formed that enables the export of the cargo out of the nucleus. Yeast cells lacking Slx9 delay export of ribosome precursors out of the nucleus. These findings imply the existence of yet-unidentified proteins like Slx9 that help Crm1 to rapidly transport diverse cargos out from the nucleus and into the cytoplasm.

(*Bischoff and Ponstingl, 1991*; *Fleischmann et al., 1991*), localizes to the nucleus, the Ran GTPase-activating protein, RanGAP1 (Rna1 in yeast), is found in the cytoplasm (*Hopper et al., 1990*; *Matunis et al., 1996*). Interactions between RanGTP and transport receptors are crucial for the directionality of nucleocytoplasmic exchange (*Nachury and Weis, 1999*). In the nucleus, RanGTP induces the release of imported cargos from importins (*Rexach and Blobel, 1995*; *Görlich et al., 1996*). In addition, RanGTP promotes the interaction of cargos with exportins for their transport to the cytoplasm (*Fornerod et al., 1997*; *Kutay et al., 1997*; *Stade et al., 1997*; *Solsbacher et al., 1998*).

In budding yeast, ribosome assembly accounts for a major proportion of the nucleocytoplasmic transport (*Rout et al., 1997*; *Schlenstedt et al., 1997*; *Sydorskyy et al., 2003*; *Kressler et al., 2012*; *Schütz et al., 2014*). mRNAs encoding ribosomal proteins (r-proteins) are exported into the cytoplasm. Newly synthesized r-proteins are imported into the nucleus and then targeted to the nucleolus for incorporation into nascent pre-ribosomes (*Schütz et al., 2014*). Additionally, >300 transiently interacting non-ribosomal assembly factors aid the construction and maturation of ribosomes (*Bassler et al., 2001*; *Dragon et al., 2002*; *Grandi et al., 2002*; *Schäfer et al., 2003*; *Gerhardy et al., 2014*). Correctly assembled pre-ribosomal particles are transported through NPCs into the cytoplasm (*Tschochner and Hurt, 2003*; *Panse and Johnson, 2010*). In addition to other cargos, it is estimated that each yeast NPC facilitates the export of ~25 pre-ribosomal particles every minute (*Warner, 1999*). Transporting pre-ribosomal cargos from the nucleus through the NPC into the cytoplasm, therefore, represents a major task for the export machinery.

The Ran-cycle-dependent exportin Crm1 plays an essential role in exporting pre-ribosomal particles to the cytoplasm (*Hurt et al., 1999*; *Moy and Silver, 1999*; *Gadal et al., 2001*; *Johnson et al., 2002*). Crm1 recognizes and directly binds leucine-rich nuclear export signals (NESs) on cargos in the presence of RanGTP to form a Crm1-export complex (*Fornerod et al., 1997*). Although an essential NES-containing adaptor, Nmd3, has been identified for 60S pre-ribosome export (*Ho and Johnson, 1999*; *Ho et al., 2000*; *Gadal et al., 2001*), a similar adaptor to recruit Crm1 to the 40S pre-ribosome remains elusive. It has been suggested that multiple NES-containing adaptors such as Ltv1 and Rio2 recruit Crm1 (*Seiser et al., 2006*; *Zemp et al., 2009*; *Merwin et al., 2014*), thereby guaranteeing efficient 40S pre-ribosome export. The essential mRNA and 60S pre-ribosome transport receptor Mex67-Mtr2, which does not directly utilize the RanGTP gradient, also facilitates nuclear

export of the 40S pre-ribosomal cargo (*Faza et al., 2012*). Despite the identification of several components of the export machinery, assembly steps and mechanisms that prepare the pre-ribosomal cargo for transport through the NPC remain largely unexplored.

To initiate export, Crm1 must cooperatively bind RanGTP and its NES-containing cargo in the nucleus, to form a trimeric export complex (*Petosa et al., 2004*; *Dong et al., 2009*; *Monecke et al., 2013*). NESs have evolved to maintain relatively low affinity to Crm1 to avoid defects in disassembly of the export complex in the cytoplasm (*Engelsma et al., 2004*; *Kutay and Güttinger, 2005*). Mechanisms that promote Crm1-export complex assembly and thereby ensure export of the NES-containing cargo at a reasonable rate remain poorly understood.

Here, we identify yeast Slx9 as a new type of RanGTP-binding protein that promotes assembly of a Crm1-export complex on the 40S pre-ribosome-associated NES-containing adaptor Rio2. Our data raise the possibility of a yet-unidentified family of RanGTP-binding proteins that act as scaffolds to optimally present RanGTP and NES-containing cargos to Crm1, orchestrating a non-cooperative stepwise assembly that drives fast and efficient Crm1-mediated export.

## Results

### *slx9-1* causes defects in 40S pre-ribosome nuclear export

Slx9 is a 24-kDa basic protein that co-enriches with pre-ribosomal particles in the 40S maturation pathway (*Gavin et al., 2002*; *Faza et al., 2012*) and is required for efficient nuclear export of 40S pre-ribosomes (*Li et al., 2009*; *Faza et al., 2012*). However, the precise contribution of Slx9 to 40S pre-ribosome export has remained unclear. To investigate the function of yeast Slx9, we generated *slx9* variants by random mutagenesis and analyzed the growth of the resulting strains at different temperatures. One allele, *slx9L108P*, hereafter termed *slx9-1*, caused slow growth at temperatures between 20°C and 30°C, indistinguishable from *slx9∆* cells (*Figure 1A*, top panel). Like *slx9∆*, *slx9-1* cells were not impaired in growth at 37°C (*Figure 1A*). Western analysis of whole cell lysates revealed that Slx9 and Slx9-1 were present at similar levels (*Figure 1A*, bottom panel), indicating that impaired growth of the *slx9-1* strain is not due to reduced levels of the mutant protein. As previously observed, Slx9-GFP localized primarily to the nucleolus, where it co-localized with the nucleolar marker Gar1-mCherry, as well as to the nucleoplasm (*Faza et al., 2012* and *Figure 1B*). Slx9-1-GFP displayed an identical localization (*Figure 1B*), indicating that the mutant protein is correctly targeted to the nucleolus and nucleoplasm. Slx9 maximally co-enriched with Enp1-TAP that purifies both the 90S and 40S pre-ribosomes (*Faza et al., 2012*). A similar purification from *slx9-1* cells revealed that Enp1-TAP co-enriched at least as much Slx9-1 mutant protein as Slx9 (*Figure 1C*). Together, these data show that Slx9-1 is correctly expressed, localized, and recruited to 40S pre-ribosomes.

Previous studies showed that *slx9∆* cells accumulate the small subunit reporter uS5-GFP (yeast Rps2, nomenclature according to *Ban et al., 2014*) and 20S pre-rRNA in the nucleoplasm (*Li et al., 2009*; *Faza et al., 2012*), indicating a defect in 40S pre-ribosome export. Using these reporters, we tested whether *slx9-1* cells have defects in 40S pre-ribosome export. Whereas WT cells displayed cytoplasmic uS5-GFP localization, *slx9-1* cells showed a strong nuclear accumulation of this reporter, similar to that observed in *slx9∆* cells (*Faza et al., 2012* and *Figure 1D*, top panel). As expected, fluorescence in situ hybridization (FISH) of 20S pre-rRNA in WT cells showed a strong nucleolar Cy3-ITS1 signal (red) with virtually no nucleoplasmic staining. In contrast, *slx9-1* cells displayed a nucleoplasmic signal of Cy3-ITS1 localization, which co-localized with the DAPI signal (*Figure 1D*, bottom panel). These data indicate that *slx9-1* cells, like *slx9∆* cells (*Faza et al., 2012*), are impaired in 40S pre-ribosome export. Therefore, we conclude that Slx9-1 is recruited to the 40S pre-ribosome but fails to fulfill its function in nuclear export of the pre-ribosomal cargo.

### Slx9 is a shuttling RanGTP-binding protein

Mutations in *MEX67* and *MTR2* (*mex67∆loop* and *mtr2∆loop116-137*), which encode the essential transport receptor Mex67-Mtr2, are synthetically lethal when combined with the *slx9∆* mutant (*Faza et al., 2012*). In addition, we found that *slx9∆* displayed a synthetic growth defect with a strain expressing Rrp12-GFP (*Figure 2A*). Rrp12 is a 40S pre-ribosome export factor that directly interacts with FG-rich nucleoporins (*Oeffinger et al., 2004*). Based on these genetic interactions, we asked whether Slx9 functions as a novel export factor for the 40S pre-ribosome. A salient feature of an export factor is that it rapidly shuttles between the nucleus and the cytoplasm. To test this, we

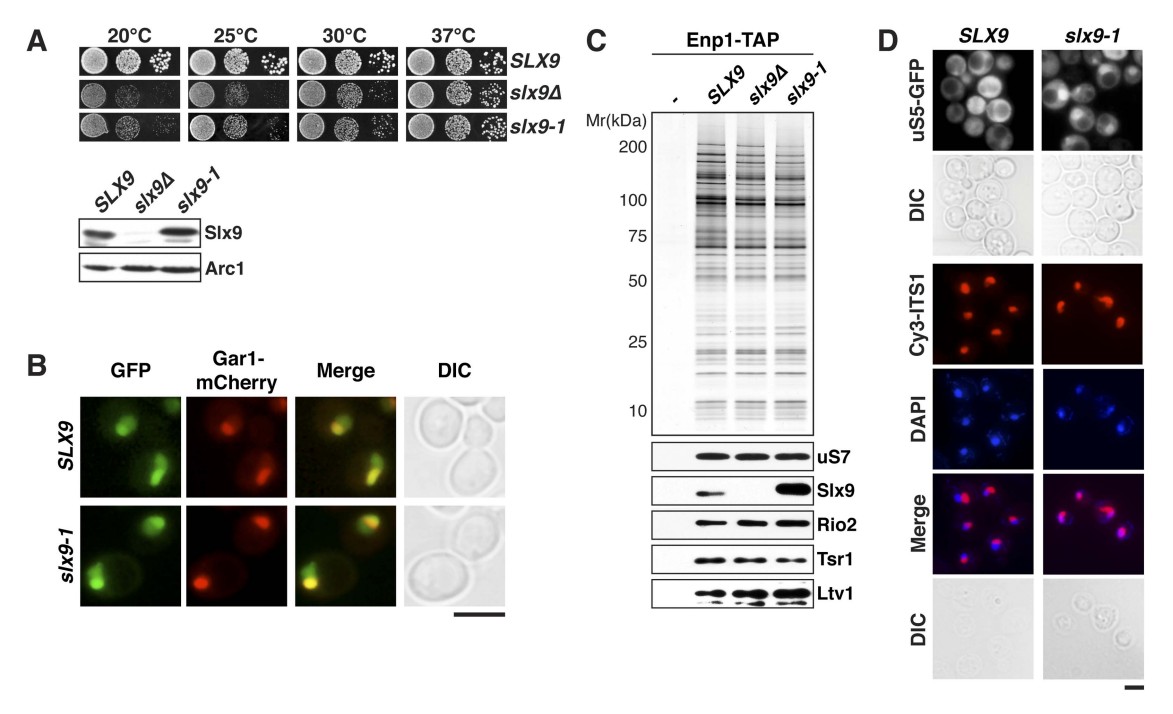

**Figure 1**. *slx9-1* phenocopies the *slx9Δ* mutation. (**A**) The *slx9-1* allele does not complement the slow growth of *slx9Δ* cells. Top: *SLX9*, *slx9Δ*, and *slx9-1* cells were spotted in 10-fold dilutions on SD-plates and grown at the indicated temperatures for 3–6 days. Bottom: Slx9 protein levels from whole cell extracts derived from the indicated strains were determined by Western analysis using antibodies directed against Slx9. Levels of the protein Arc1 served as a loading control. (**B**) Slx9-1 localizes to the nucleolus/nucleoplasm. Cells expressing Gar1-mCherry and Slx9-GFP or Slx9-1-GFP were grown until mid-log phase. Localization of the indicated fusion proteins was analyzed by fluorescence microscopy. Gar1-mCherry served as a nucleolar marker. Scale bar = 5 μm. (**C**) Slx9-1 is recruited to the early 40S pre-ribosome. Enp1-TAP was isolated by tandem affinity purification (TAP) from the indicated strains. Calmodulin-eluates were separated on a 4–12% gradient gel and analyzed by either silver staining or Western using the indicated antibodies. The ribosomal protein uS7 served as a loading control. (**D**) *slx9-1* cells are impaired in nuclear export of 40S pre-ribosomes. Top: localization of uS5-GFP was monitored by fluorescence microscopy. Bottom: localization of 20S pre-rRNA was analyzed by FISH using a Cy3-labeled oligonucleotide complementary to the 5′ portion of ITS1 (red). Nuclear and mitochondrial DNA was stained by DAPI (blue). Scale bar = 5 μm.

employed the established heterokaryon assay (*Altvater et al., 2014*). WT cells expressing Slx9-GFP were mated to *kar1-1* cells, which are deficient in nuclear fusion after cell conjugation, leading to heterokaryon formation. In order to distinguish between the two nuclei, *kar1-1* cells also contained Nup82-mCherry as a marker for nuclear pores. As controls, we used the shuttling 40S assembly factor Enp1 and the non-shuttling nucleolar protein Gar1 fused to GFP. Whereas Gar1-GFP was never seen in the nucleus of *kar1-1* cells (red signal), both Enp1-GFP and Slx9-GFP localized to both nuclei (*Figure 2B*). These data are consistent with the shuttling of Slx9 between the nuclear and the cytoplasmic compartments.

We wondered whether Slx9 functions directly in 40S pre-ribosome export as a NES-containing adaptor for the exportin Xpo1 (hereafter termed Crm1). We, therefore, investigated whether Slx9 binds Crm1 in the presence of Gsp1Q71L (*Maurer et al., 2001*) in vitro (equivalent to the human RanQ69L GTP-stabilized mutant, hereafter termed Ran$^{QL}$GTP; *Bischoff et al., 1994*). The C-terminal domain of Ssb1 (Ssb1C) that contains a functional NES (*Shulga et al., 1999*) served as a positive control. Unlike Ssb1C (*Maurer et al., 2001* and *Figure 2C*, lane 8), Slx9 was unable to form a trimeric export complex with Crm1 and Ran$^{QL}$GTP (*Figure 2C*, lane 4). Surprisingly, these studies revealed that Slx9 directly bound Ran$^{QL}$GTP (*Figure 2C*, lane 2 and 4). Therefore, although Slx9 does not contain a functional NES, it is a Ran-binding protein.

Since Slx9 is a shuttling protein, we tested whether Slx9 interacts with both RanGTP and RanGDP in vitro. As controls, we used Ntf2, the import factor for RanGDP (*Ribbeck et al., 1998*; *Smith et al., 1998* and *Figure 2D*, lane 8) and the yeast RanBP1 homolog Yrb1 that binds to RanGTP

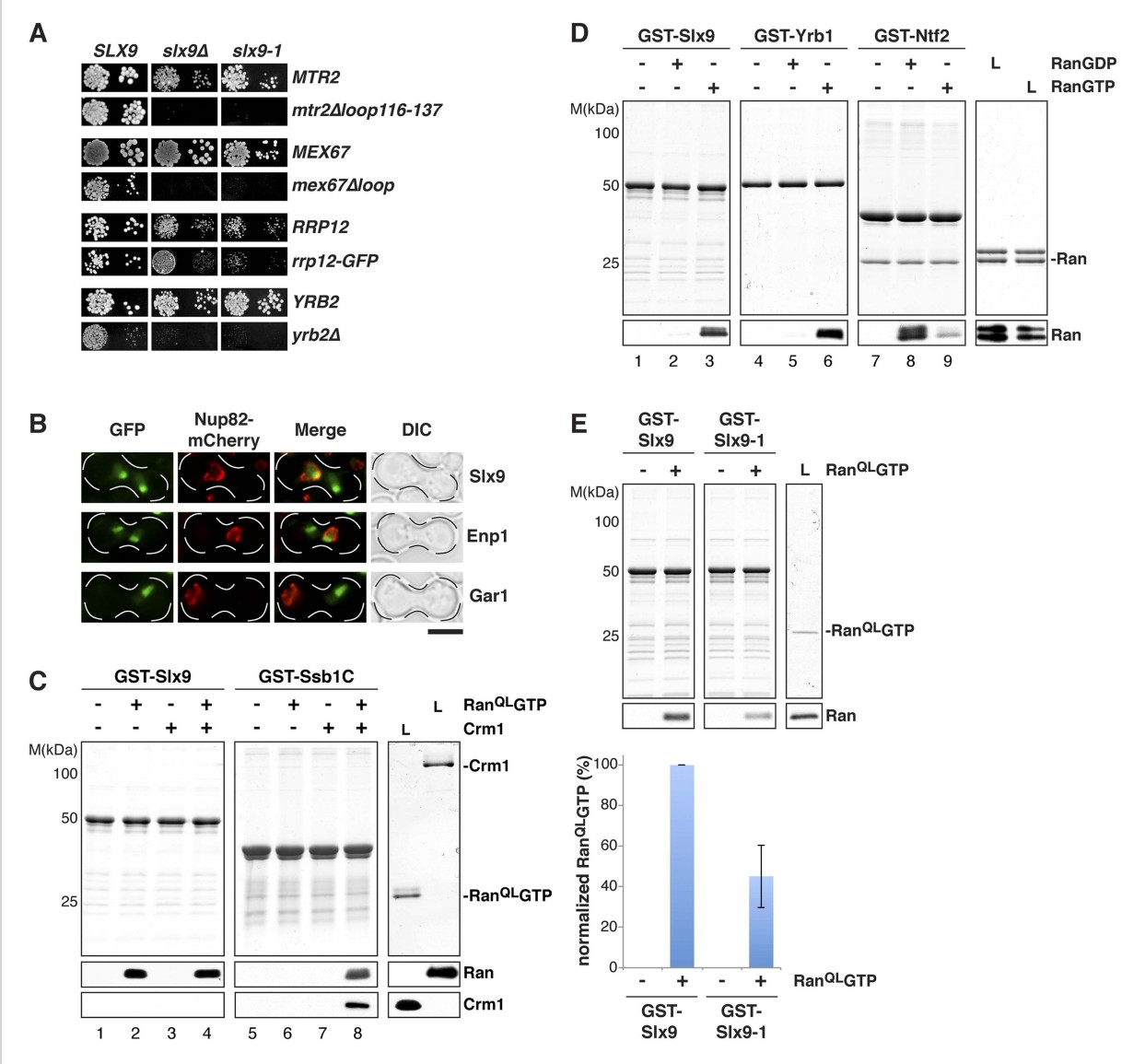

**Figure 2**. Slx9 is a RanGTP binding protein. (**A**) *slx9-1* genetically interacts with factors involved in 40S pre-ribosome export. *slx9-1* is synthetically lethal with *mex67Δloop*, *mtr2Δloop116-137*, or *yrb2Δ* and strongly synthetically enhanced with *rrp12-GFP*. Strains containing the indicated WT and mutant alleles were spotted in 10-fold serial dilutions on 5-FOA-SD or SD and grown at 20–30°C for 3–6 days. (**B**) Slx9 shuttles between the nucleus and the cytoplasm. Cells expressing Enp1-GFP, Gar1-GFP, or Slx9-GFP were mated with *kar1-1* cells expressing Nup82-mCherry. The resulting heterokaryons were analyzed by fluorescence microscopy. Scale bar = 5 μm. (**C**) Slx9 directly binds to RanGTP. GST-Slx9 or GST-Ssb1C was immobilized on GSH-Sepharose before incubating with either buffer alone or buffer containing 2 μM His$_6$-Ran$^{QL}$GTP, 50 nM Crm1-His$_6$ or 2 μM His$_6$-Ran$^{QL}$GTP, and 50 nM Crm1-His$_6$. After washing, bound proteins were eluted in LDS sample buffer, separated by SDS-PAGE and visualized by Coomassie staining or Western blotting using the indicated antibodies. L = input. (**D**) Slx9 specifically interacts with the GTP-bound form of Ran. GST-Slx9, GST-Yrb1, or GST-Ntf2 was immobilized on GSH-Sepharose and incubated with buffer alone or 2 μM His$_6$-Ran loaded with GDP or GTP. Analysis of the eluted proteins was carried out as described in (**C**). L = input. (**E**) Slx9-1 binding to RanGTP is impaired. Top: GST-Slx9 or GST-Slx9-1 immobilized on GSH-Sepharose was incubated with buffer alone or 2 μM His$_6$-Ran$^{QL}$GTP. Analysis of the eluted proteins was carried out as described in (**C**). L = input. Bottom: bar graph depicts the bound His$_6$-Ran$^{QL}$GTP Western blot signal normalized to GST-Slx9 and GST-Slx9-1 levels, respectively. Four independent experiments were performed and Western blots were quantified by software ImageJ (Version 1.44o). Error bars (S.D.) are indicated.

(*Schlenstedt et al., 1995* and *Figure 2D*, lane 6). We found that, like Yrb1, Slx9 interacted exclusively with RanGTP (*Figure 2D*, lanes 2 and 3). Based on these data, we conclude that Slx9 is a shuttling RanGTP-binding protein.

## Slx9-1 is impaired in binding Ran$^{QL}$GTP

The *slx9-1* mutant did not rescue the slow growth and impaired 40S pre-ribosome export of *slx9Δ* cells (*Figure 1A*). Furthermore, like *slx9Δ* cells, *slx9-1* cells genetically interacted with *mex67* and *mtr2* mutants (*mex67Δloop* and *mtr2Δ116-137*) and *rrp12-GFP* (*Figure 2A*). These findings prompted us to test whether Slx9-1 binds to Ran$^{QL}$GTP in vitro, using the assay described above. We found a decrease of approximately 50% in the levels of Ran$^{QL}$GTP bound to Slx9-1 as compared to Slx9 (*Figure 2E*). Based on these data, we conclude that Slx9-1 is modestly impaired in binding Ran$^{QL}$GTP.

## The basic patch of RanGTP contributes to Slx9 binding

A conserved basic patch on Ran is involved in the interaction with known Ran-binding proteins (*Nilsson et al., 2001*). Based on homology to human Ran, arginine 142, and lysine 143 in yeast Ran were mutated to alanine (Ran$^{QL}$GTP$^{RKAA}$) or glutamate (Ran$^{QL}$GTP$^{RKEE}$) and the contribution of this basic patch to Slx9:RanGTP complex formation was analyzed in vitro. In agreement with previous studies (*Nilsson et al., 2001*), these Ran$^{QL}$GTP mutants bound weakly to the importin β-like transport receptor, Kap123 (*Figure 3A*, compare lane 10 with lanes 11 and 12), and interacted more strongly with the RanBP1 homolog Yrb1 (*Figure 3A*, compare lane 6 with lanes 7 and 8). Pull down studies of Slx9 and these Ran mutants showed that the interactions between Slx9 and these two Ran mutants were impaired, with the charge reversal mutant having a more severe effect than the alanine mutant (*Figure 3A*, compare lane 2 with lanes 3 and 4). Altogether, these results suggest that, similar to Kap123, Slx9 binding to Ran$^{QL}$GTP involves the basic patch.

## The acidic C-terminal tail of Ran reduces Slx9-binding

The C-terminal acidic tail of Ran (-DEDDAL) plays a crucial role in the interaction with small RanGTP-binding proteins such as Yrb1. As expected, Ran$^{QL}$GTP lacking the C-terminal acidic tail (RanΔC$^{QL}$GTP) failed to interact with Yrb1 in vitro (*Maurer et al., 2001* and *Figure 3B*, lane 6). We tested whether the acidic tail contributes to the interaction between RanGTP and Slx9. In contrast to Yrb1, RanΔC$^{QL}$GTP bound stronger to Slx9 compared to Ran$^{QL}$GTP (*Figure 3B*, compare lane 2 and 3). This enhanced interaction was specific, since Kap123 bound RanΔC$^{QL}$GTP and Ran$^{QL}$GTP to a similar extent (*Figure 3B*, compare lanes 8 and 9). These data suggest that the C-terminal acidic tail of Ran negatively regulates RanGTP:Slx9 interactions.

## *SLX9* genetically interacts with the RanGTP- and Crm1-binding protein Yrb2

Cells lacking the RanGTP- and Crm1-binding protein Yrb2 (*yrb2Δ*) exhibit strong nucleoplasmic accumulation of uS5-GFP and 20S pre-rRNA as well as reduced abundance of 40S subunits (*Moy and Silver, 2002*; *Altvater et al., 2012*). In addition, *mex67Δloop* and *mtr2Δloop116-137* are synthetically lethal with *yrb2Δ* (*Faza et al., 2012*). These findings led us to test whether *SLX9* genetically interacts with *YRB2*. Both *slx9Δ* and *slx9-1* were synthetically lethal with *yrb2Δ* (*Figure 2A*), suggesting that Slx9 and Yrb2 functionally overlap to ensure proper nuclear export of 40S pre-ribosomes.

## Slx9 binds the NES-containing 40S pre-ribosomal adaptor Rio2

Yrb2 and its human homolog, RanBP3, stimulate the assembly of Crm1-export complexes on certain NES-containing cargos by cooperatively binding Crm1 and RanGTP (*Englmeier et al., 2001*; *Lindsay et al., 2001*; *Koyama et al., 2014*). The strong genetic interaction between *SLX9* and *YRB2* raised the possibility that Slx9 also functions in Crm1-complex assembly. However, our interaction studies showed that Slx9 binds RanGTP, but not Crm1 (*Figure 2C*, lane 4). We, therefore, wondered whether Slx9 instead facilitates the assembly of a Crm1-export complex by bringing together RanGTP and the NES-containing adaptor.

Two 40S pre-ribosome-associated factors, hLtv1 and hRio2, have been shown to bind Crm1 in the presence of RanGTP (*Zemp et al., 2009*). In agreement with these studies, we found that yeast Ltv1 and yeast Rio2 formed trimeric complexes with Crm1 and Ran$^{QL}$GTP via a cooperative mechanism in vitro (*Figure 4A*, lanes 4 and 12). Moreover, the C-terminal regions of these proteins are predicted to contain a NES (*Zemp et al., 2009*; *Merwin et al., 2014* and *Figure 4A*, top panel), and indeed, Rio2 and Ltv1 mutant proteins lacking NESs were unable to form trimeric export complexes (*Figure 4A*,

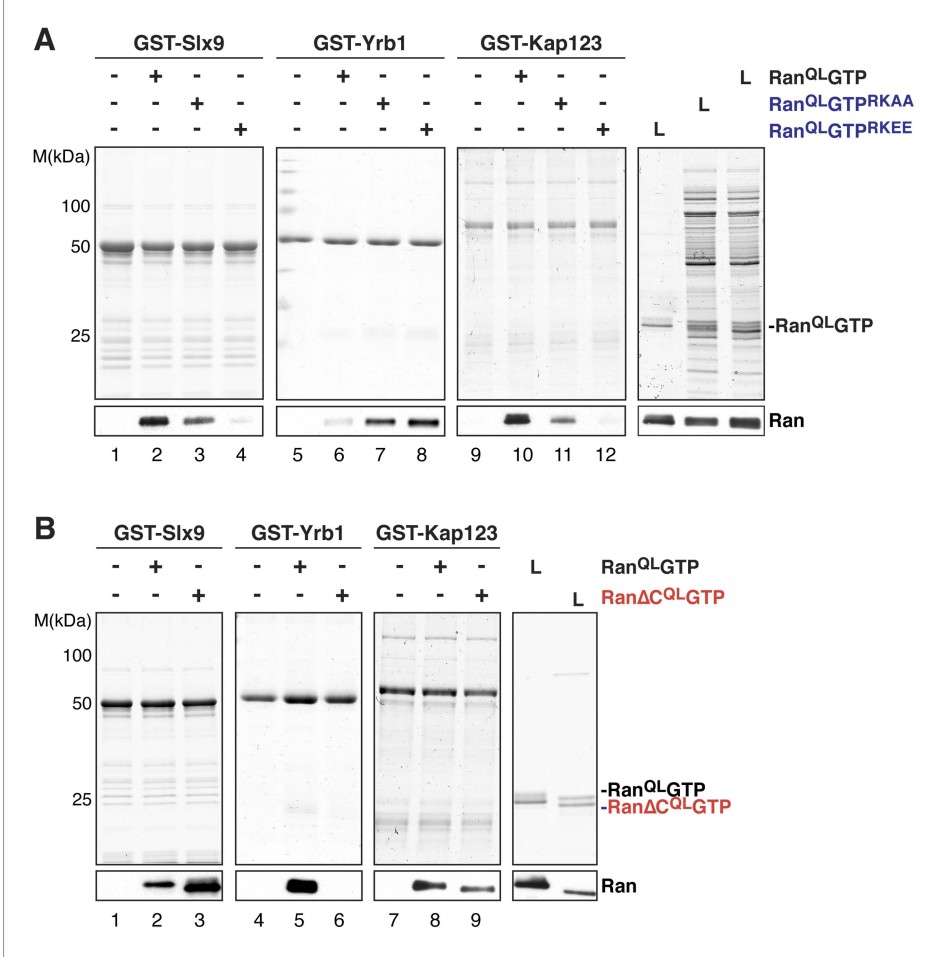

**Figure 3**. The basic patch and acidic tail of Ran modulates interactions with Slx9. (**A**) The basic patch of $Ran_{QL}GTP$ contributes to Slx9 binding. GST-Slx9, GST-Yrb1, or GST-Kap123 immobilized on GSH-Sepharose was incubated with buffer alone or 2 μM Ran ($His_6$-$Ran^{QL}GTP$, $His_6$-$Ran^{QL}GTP^{RKAA}$, or $His_6$-$Ran^{QL}GTP^{RKEE}$). After washing, bound proteins were eluted in LDS sample buffer, separated by SDS-PAGE and visualized by Coomassie staining or Western blotting using the indicated antibody. L = input. (**B**) The acidic tail of $Ran^{QL}GTP$ negatively regulates interactions with Slx9. GST-Slx9, GST-Yrb1, or GST-Kap123 immobilized on GSH-Sepharose was incubated with buffer alone or 2 μM Ran ($His_6$-$Ran^{QL}GTP$ or $His_6$-$Ran\Delta C^{QL}GTP$). Analysis of the eluted proteins was carried out as described in (**A**). L = input.

lanes 8 and 16). Western analyses revealed that Rio2, but not Ltv1, interacted weakly with $Ran^{QL}GTP$, independent of Crm1 (**Figure 4A**, lane 2).

Ltv1 and Rio2 co-enrich with the Enp1-TAP particle that also maximally co-purifies Slx9 (**Figure 1C**; **Schütz et al., 2014**) suggesting that these NES-containing proteins might interact with Slx9. To test this, we incubated immobilized GST-Rio2 or GST-Ltv1 with Slx9. Slx9 directly bound to Rio2 but not to Ltv1 (**Figure 4B**, left panel). Conversely, GST-Slx9 interacted with Rio2 but not with the essential 60S pre-ribosome-associated NES-containing adaptor Nmd3 (**Figure 4B**, right panel). Moreover, the GST-Rio2:Slx9 complex efficiently recruited $Ran^{QL}GTP$ (**Figure 5A**). The level of recruitment was not affected by the prior presence or absence of Slx9, since GST-Rio2 saturated with Slx9 recruited similar amounts of $Ran^{QL}GTP$ (**Figure 5B**, compare top and bottom panels). Also, the levels of $Ran^{QL}GTP$ recruitment to GST-Slx9 were not affected by the presence or absence of Rio2 (**Figure 5C**, compare top and bottom panels). Altogether, these data suggest that Rio2 binds Slx9 and $Ran^{QL}GTP$ using distinct surfaces.

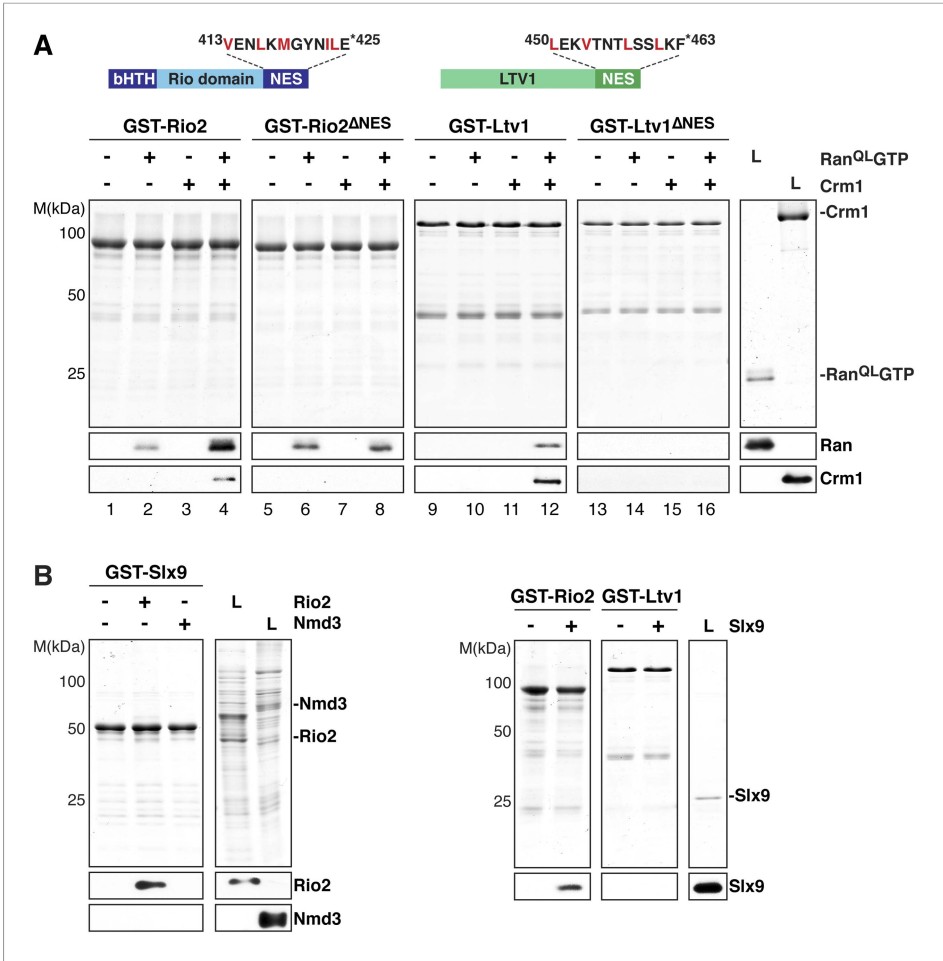

**Figure 4**. Slx9 directly binds the 40S pre-ribosome nuclear export signal (NES)-containing adaptor Rio2 and Ran$^{QL}$GTP. (**A**) Rio2 and Ltv1 export complex formation requires their C-terminal NESs. Top: the positions of the Rio2 and Ltv1 NESs are shown. Hydrophobic residues in these NESs are highlighted in red. Bottom: GST-Rio2, GST-Rio2$^{\Delta NES}$, GST-Ltv1, or GST-Ltv1$^{\Delta NES}$ was immobilized on GSH-Sepharose, and complex formation was analyzed as in *Figure 2C*. L = input. (**B**) Slx9 directly interacts with Rio2. Immobilized GST-Rio2 or GST-Ltv1 was incubated with buffer alone or 0.5 μM Slx9. Conversely, immobilized GST-Slx9 was incubated with buffer alone or with lysate containing His$_6$-Nmd3 or His$_6$-Rio2. Analysis of the eluted proteins was carried out as described in *Figure 2C*. L = input.

## Recruitment of Crm1 to Rio2:RanGTP is stimulated by Slx9

We next investigated whether the GST-Rio2:Slx9:Ran$^{QL}$GTP complex could directly recruit Crm1. To this end, pre-formed GST-Rio2:Ran$^{QL}$GTP or GST-Rio2:Slx9:Ran$^{QL}$GTP complexes (summarized in *Figure 6A*) were incubated with buffer alone or Crm1 (*Figure 6B*, lanes 3, 4 and 7, 8). Only the GST-Rio2:Slx9:Ran$^{QL}$GTP complex efficiently recruited Crm1 (*Figure 6B*, compare lanes 4 and 8). The Crm1-recruitment to a GST-Rio2:Slx9:Ran$^{QL}$GTP complex was dependent on the NES of Rio2, since a GST-Rio2$^{\Delta NES}$:Slx9:Ran$^{QL}$GTP complex was unable to bind Crm1 (*Figure 6B*, compare lanes 8 and 10). Moreover, Crm1 recruitment was also dependent on the Ran$^{QL}$GTP bound to Rio2, since a GST-Rio2:Slx9 complex was unable to bind Crm1 (*Figure 6C*, lane 4). These studies indicate that, in order to recruit Crm1 in a non-cooperative manner, Rio2 must bind to both Slx9 and RanGTP.

We next investigated whether Slx9-1 could assemble a GST-Rio2:Slx9-1:Ran$^{QL}$GTP complex. In vitro binding studies revealed that Rio2 binds Slx9-1 (*Figure 6D*, lane 6). In comparison to the Slx9, Slx9-1 bound at least as well to Rio2 (*Figure 6D*, compare lane 2 and 6). Further, the GST-Rio2:Slx9-1 complex was also able to recruit Ran$^{QL}$GTP (*Figure 6D*, lane 7), indicating that Slx9-1 is still able to

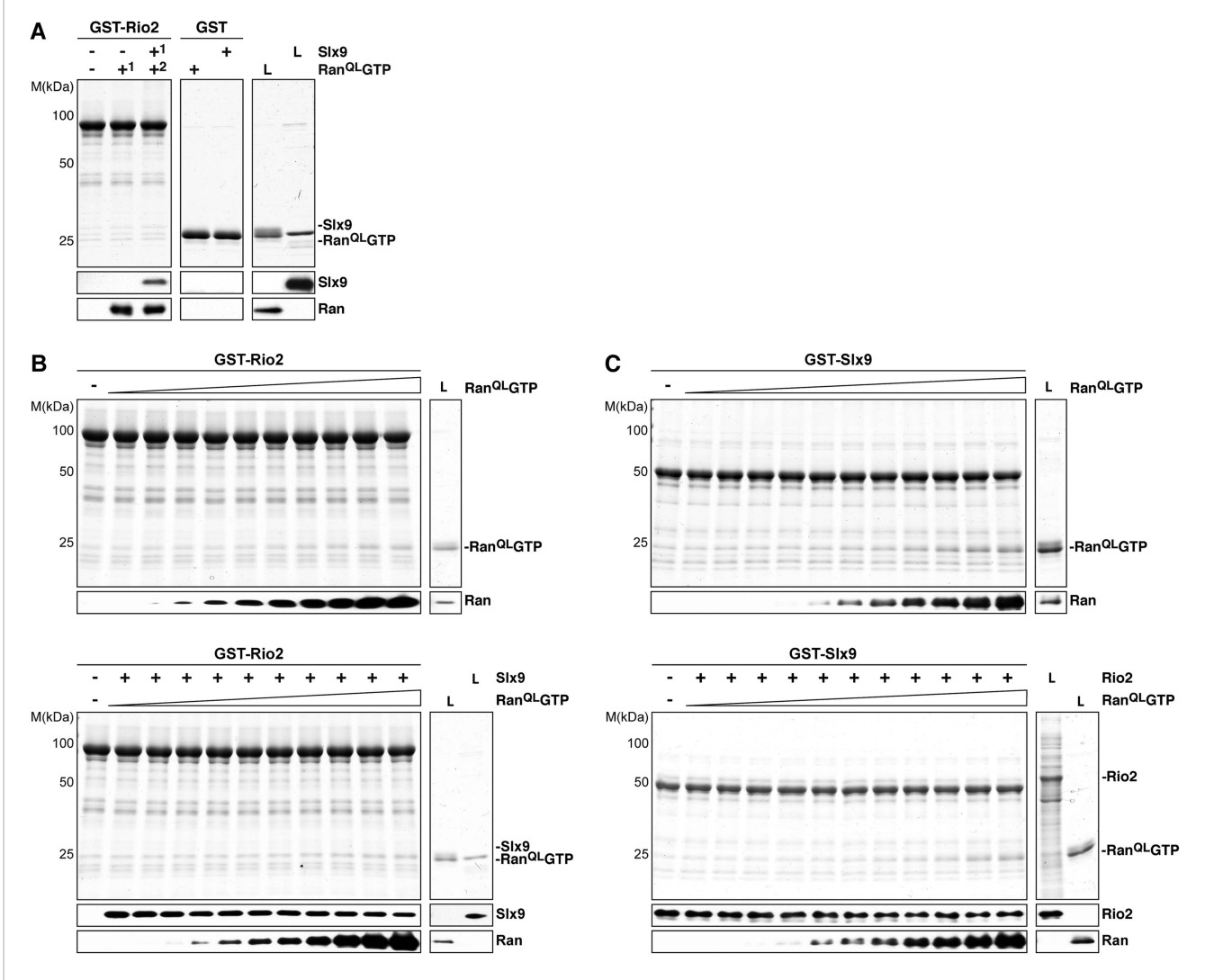

**Figure 5**. Slx9 binds to Rio2 and RanGTP using distinct binding surfaces. (**A**) GST-Rio2 was immobilized on GSH-Sepharose and incubated with buffer, 2 μM His$_6$-Ran$^{QL}$GTP, or 0.5 μM Slx9 (+[1]). After washing, the GST-Rio2:Slx9 complex was incubated with 2 μM His$_6$-Ran$^{QL}$GTP (+[2]). Analysis of the eluted proteins was carried out as described in *Figure 2C*. L = input. (**B**) RanGTP does not displace Slx9 from a preformed GST-Rio2:Slx9 complex. Top: immobilized GST-Rio2 was incubated with buffer or increasing concentrations of His$_6$-Ran$^{QL}$GTP (62.5 nM–32 μM). Bottom: immobilized GST-Rio2 was incubated with either buffer or 1 μM Slx9. The unbound Slx9 was washed away, and the resulting GST-Rio2:Slx9 complex was incubated with increasing concentrations of His$_6$-Ran$^{QL}$GTP (62.5 nM–32 μM). Analysis of the eluted proteins was carried out as described in *Figure 2C*. L = input. (**C**) Ran$^{QL}$GTP does not displace Rio2 from a preformed GST-Slx9:Rio2 complex. Top: immobilized GST-Slx9 was incubated with buffer or increasing concentrations of His$_6$-Ran$^{QL}$GTP (62.5 nM–32 μM). Bottom: immobilized GST-Slx9 was incubated with excess of Rio2. The unbound Rio2 was washed away, and the resulting complex GST-Slx9:Rio2 complex was incubated with increasing concentrations of His$_6$-Ran$^{QL}$GTP (62.5 nM–32 μM). Analysis of the eluted proteins was carried out as described in *Figure 2C*. L = input.

assemble a GST-Rio2:Slx9-1:Ran$^{QL}$GTP complex. However, this complex was impaired in loading Crm1 (*Figure 6D*, compare lane 5 and 9). Therefore, we suggest that Slx9 promotes assembly of a Crm1-export complex on Rio2, and that this function relies on a proper interaction between Slx9 and RanGTP.

## Slx9 acts as a scaffold to load Crm1 onto the Rio2-NES

Crm1 recognizes and binds cargos that contain diverse leucine-rich NESs. Structural analyses of the RanGTP:Crm1 complex bound to prototypic NESs suggest that any peptide can function as a NES as long as its backbone conformation permits its side chains to access the rigid hydrophobic pockets of

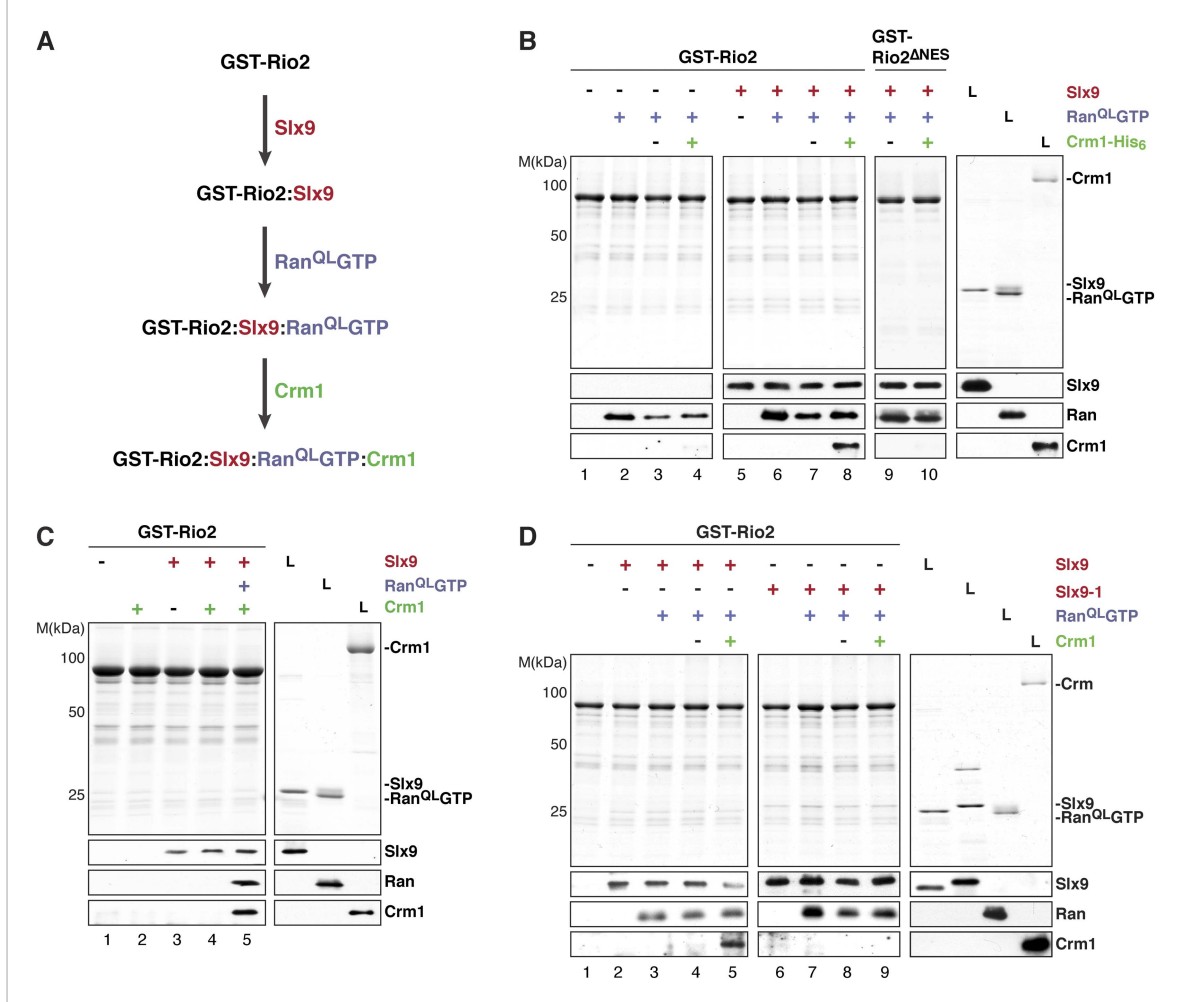

**Figure 6**. Slx9 promotes stepwise assembly of a Crm1-export complex on the NES of Rio2. (**A**) Flow chart depicting the experimental setup to assemble a Rio2:Slx9:Ran$^{QL}$GTP:Crm1 complex. Immobilized GST-Rio2 was sequentially incubated with Slx9 (red), Ran$^{QL}$GTP (purple), and Crm1 (green). Unbound protein was washed away after each incubation step. (**B**) Crm1 is recruited to the GST-Rio2:Slx9:RanGTP complex in a NES-dependent manner. Immobilized GST-Rio2 or GST-Rio2$^{\Delta NES}$ was incubated with buffer alone or 0.5 μM Slx9, followed by the stepwise addition of 0.2 μM His$_6$-Ran$^{QL}$GTP and 50 nM Crm1-His$_6$, as depicted in (**A**). After a final washing step, bound proteins were analyzed as in *Figure 2C*. L = input. (**C**) Crm1 is not recruited to the GST-Rio2:Slx9 complex. Immobilized GST-Rio2 was incubated with buffer alone or 0.5 μM Slx9, followed by addition of buffer, 50 nM Crm1-His$_6$, or the stepwise addition of 0.2 μM His$_6$-Ran$^{QL}$GTP and 50 nM Crm1-His$_6$ as depicted in (**A**). Analysis of the bound proteins was carried out as described in *Figure 2C*. L = input. (**D**) Recruitment of Crm1 to a Rio2:Slx9-1:Ran$^{QL}$GTP complex is impaired. Immobilized GST-Rio2 was incubated with buffer alone, 0.5 μM Slx9 or 0.5 μM Slx9-1, followed by the stepwise addition of 0.2 μM His$_6$-Ran$^{QL}$GTP and 50 nM Crm1-His$_6$ as depicted in (**A**). Analysis of the bound proteins was carried out as described in *Figure 2C*. L = input.

Crm1 (*Güttler et al., 2010*). To test whether conformational rigidity of the Rio2-NES is critical to recruit Crm1 in the presence of Ran$^{QL}$GTP, three consecutive residues (399-EEN-401) proximal to the NES were mutated to glycines (Rio2$^{3G}$) (*Figure 7A*, top panel). Because glycine residues lack a side chain, they allow greater conformational flexibility for the polypeptide backbone of these residues (*Ramachandran and Sasisekharan, 1968*) as well as to the neighboring NES, thus destabilizing it. We found that, like the Rio2$^{\Delta NES}$ (*Figure 4A*, lane 8), Rio2$^{3G}$ was unable to cooperatively recruit Crm1 in the presence of Ran$^{QL}$GTP in vitro (*Figure 7A*, bottom panel, lane 4). In parallel, we made a Rio2 mutant in which residues 399–401 were replaced by alanines (Rio2$^{3A}$) (*Figure 7—figure supplement 1*, top panel). Unlike Rio2$^{3G}$ (*Figure 7A*, bottom panel, lane 4), Rio2$^{3A}$ was able to efficiently cooperatively recruit Crm1 in the presence of Ran$^{QL}$GTP (*Figure 7—figure supplement 1*, bottom panel), suggesting that the glycine mutations destabilize the NES.

We next assessed the functionality of Rio2$^{\Delta NES}$ and Rio2$^{3G}$ in yeast. Both *rio2ΔNES* and *rio2$^{3G}$* rescued the lethality of the *rio2Δ* strain (*Figure 7B*). However, *rio2ΔNES* was synthetically lethal mex67Δloop and *mtr2Δ116-137* (*Figure 7B*), consistent with the model that the transport receptor Mex67-Mtr2 has a redundant function in 40S pre-ribosome nuclear export. Curiously, the *rio2$^{3G}$* allele that did not recruit Crm1 in the presence of Ran$^{QL}$GTP in vitro (*Figure 7A*, bottom panel, lane 4), rescued the synthetic lethality (*Figure 7B*), indicating that Rio2$^{3G}$ is still functional in vivo. Importantly, neither *rio2* alleles (*rio2ΔNES* and *rio2$^{3G}$*) were synthetic lethal when combined with *slx9Δ* and *slx9-1* mutant strains (*Figure 7—figure supplement 2A*).

These genetic interactions led us to ask whether Slx9 could stabilize the NES conformer of Rio2$^{3G}$ to facilitate Crm1 recruitment. To test this, a GST-Rio2$^{3G}$:Slx9:Ran$^{QL}$GTP complex was incubated with Crm1. Remarkably, this complex was able to recruit Crm1 similar to the Rio2:Slx9:Ran$^{QL}$GTP complex (*Figure 7C*, compare lane 2 and 4). Notably, we found that the GST-Rio2$^{3G}$:Slx9-1:Ran$^{QL}$GTP complex was impaired in loading Crm1 (*Figure 7D*, compare lane 1 and 2). Altogether, these data suggest that, within the GST-Rio2$^{3G}$:Slx9:Ran$^{QL}$GTP complex, Slx9 promotes Crm1 loading by stabilizing the region surrounding the Rio2-NES.

## Strong NESs on Rio2 bypass the requirement for Slx9

Our data so far show that a specific mutation within Slx9 impaired Crm1-export complex assembly in vitro and 40S pre-ribosome export in vivo. We, therefore, wondered whether replacing the Rio2-NES with a set of strong NESs could bypass the requirement of Slx9 in 40S pre-ribosome export. The strength of a specific NES is based on its resemblance to the consensus sequence and has been shown to strongly correlate with its affinity for Crm1 in vitro (*Engelsma et al., 2004*; *Kutay and Güttinger, 2005*). To this end, we replaced the Rio2-NES with the strong NESs of Nmd3 (hereafter termed as Rio2$^{Nmd3NES}$) (*Engelsma et al., 2004*; *Kutay and Güttinger, 2005*).

Functional analyses revealed that the expression of Rio2$^{Nmd3NES}$ complemented the lethality of the *rio2Δ* strain. Moreover, *rio2-nmd3NES* was not synthetic lethal with *mex67Δloop* and *mtr2Δ116-137* (*Figure 8A*). Since Rio2$^{Nmd3NES}$ bound to Crm1 in the presence of RanGTP (*Figure 8B*, lane 4), we assessed whether Rio2$^{Nmd3NES}$ expression rescued the 40S pre-ribosome export defect seen in *slx9Δ* cells. >95% of *slx9Δ* cells expressing Rio2 and Rio2$^{\Delta NES}$ accumulated uS5-GFP in the nucleoplasm. However, *slx9Δ* cells expressing Rio2$^{Nmd3NES}$ did not accumulate uS5-GFP in the nucleoplasm (*Figure 8C*), indicating no apparent impairment in 40S pre-ribosome export. A Rio2 variant containing only the first NES of Nmd3 (*rio2-nmd3NESΔ1*) was unable to rescue the 40S pre-ribosome export defect of *slx9Δ* cells (*Figure 8C*), suggesting that both NESs are required to bypass Slx9 function in 40S pre-ribosome export. Notably, the expression of Rio2$^{Nmd3NES}$ in *yrb2Δ* cells did not rescue the nucleoplasmic accumulation of uS5-GFP (*Figure 8D*), indicating that the heterologous NESs specifically bypasses Slx9 function but not other steps that drive 40S pre-ribosome export.

## Discussion

In order to mediate nuclear export of NES-containing cargos, Crm1 must form trimeric complexes with these cargos and RanGTP. To allow efficient cargo release in the cytoplasm, NESs have evolved to have low affinity for Crm1 (*Engelsma et al., 2004*; *Kutay and Güttinger, 2005*). This low affinity is expected to reduce the efficiency of Crm1-export complex formation and slow down nuclear export (*Engelsma et al., 2004*). Despite their low affinity in vitro, NES-containing cargos are exported efficiently in vivo (*Kutay and Güttinger, 2005*). Because they are amongst the most abundant export cargos in a eukaryotic cell and require rapid, sustained rates of export; pre-ribosomal particles are particularly sensitive to the interactions between the NES and Crm1 (*Zemp and Kutay, 2007*; *Panse and Johnson, 2010*). Here, we identify Slx9 as a RanGTP-binding protein that stabilizes the interaction between RanGTP, Rio2, and Crm1 without promoting cooperative binding between these proteins. Instead, Slx9 overcomes the low affinity between the NES and Crm1 and drives stepwise assembly of a Crm1-export complex on the NES-containing adaptor Rio2 of the 40S pre-ribosomal cargo.

## Slx9 is a new type of RanGTP-binding protein required for 40S pre-ribosome export

Slx9 was first proposed to function in ribosome biogenesis pathway through promoting ITS1 processing, which is required to separate pre-rRNAs of the large and small pre-ribosomal subunits

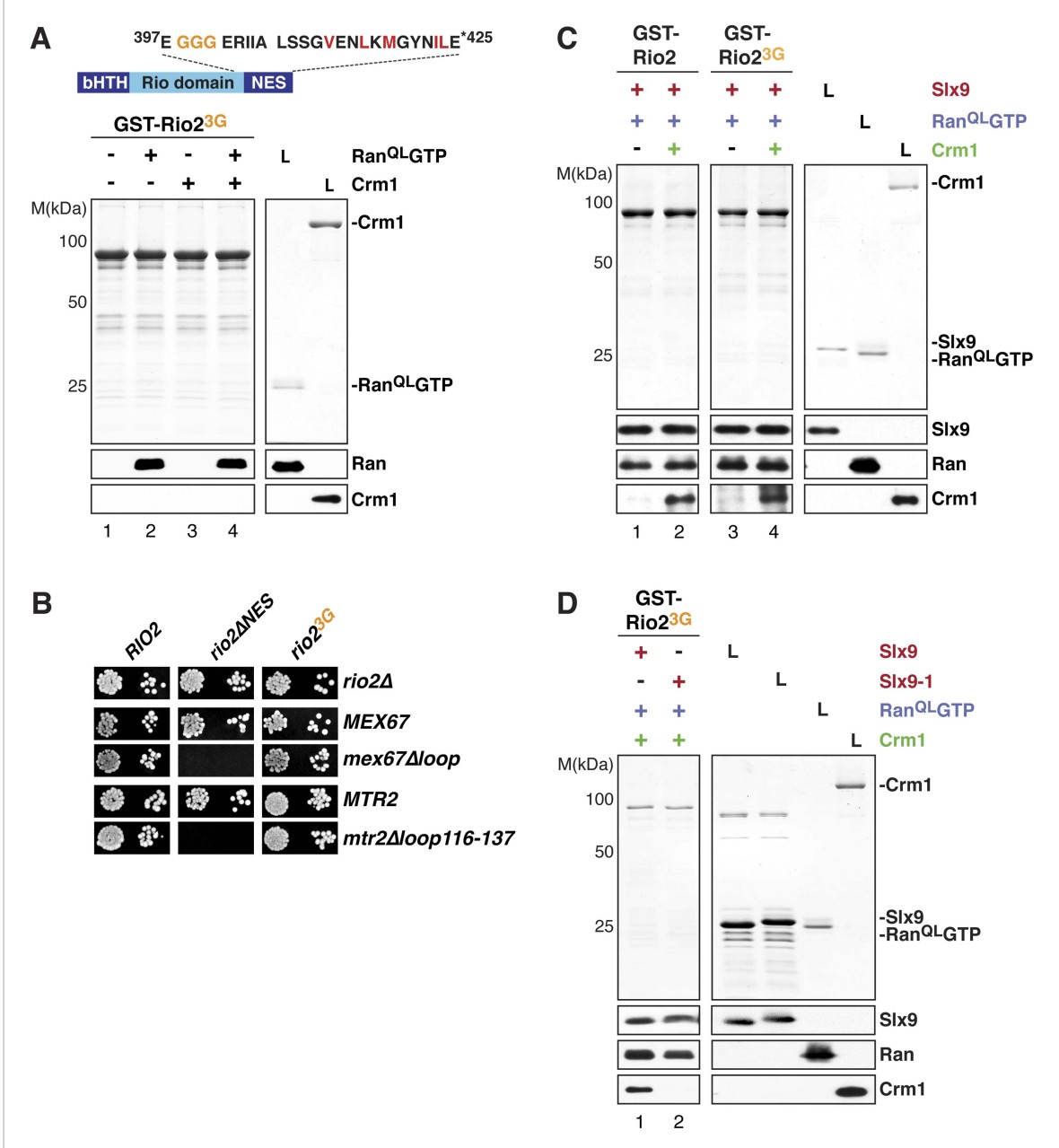

**Figure 7**. Slx9 provides a scaffold to load Crm1 onto Rio2-NES. (**A**) Rio2[3G] does not interact with Crm1 in the presence of RanGTP. Top: schematic depicts the positions of mutations proximal to the NES (399-EEN-401-GGG) in the Rio2[3G]. Hydrophobic amino acids of the NES are red and mutated amino acids are orange. Bottom: GST-Rio2[3G] was immobilized on GSH-Sepharose and binding reactions were carried out and analyzed as in *Figure 2C*. L = input. (**B**) *rio2ΔNES*, but not *rio2[3G]*, is synthetically lethal with *mex67Δloop* and *mtr2Δloop116-137*. Strains were spotted in 10-fold serial dilutions on 5-FOA (SD) plates and grown at 30°C for 2–4 days. (**C**) Slx9 restores Crm1 binding to the Rio2[3G]:Slx9:Ran[QL]GTP complex. GST-Rio2:Slx9:Ran[QL]GTP or GST-Rio2[3G]:Slx9:Ran[QL]GTP was incubated with buffer alone or 50 nM Crm1-His[6]. Bound proteins were analyzed as in *Figure 2C*. L = input. (**D**) Crm1 is impaired in binding a Rio2[3G]:Slx9-1:Ran[QL]GTP complex. Immobilized GST-Rio2[3G] was incubated with buffer alone, 0.5 μM Slx9 or 0.5 μM Slx9-1, followed by the stepwise addition of 0.2 μM His[6]-Ran[QL]GTP and 50 nM Crm1-His[6] as depicted in (**A**). Analysis of the bound proteins was carried out as described in *Figure 2C*. L = input.

The following figure supplements are available for figure 7:

**Figure supplement 1**. The 'flexibility' of the NES region in Rio2 contributes to its interaction with Crm1 in the presence of RanGTP.

**Figure supplement 2**. Genetic interactions between Rio2 alleles and RanGTP-binding proteins Slx9 and Yrb2.

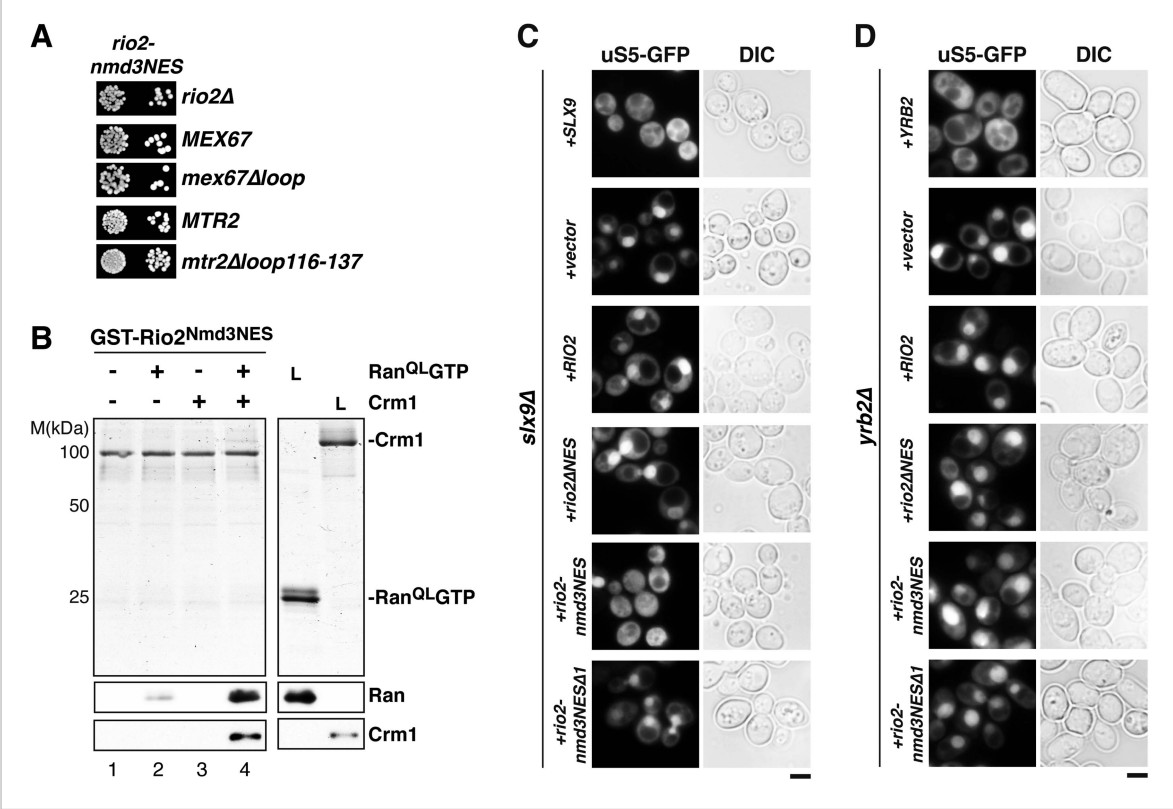

**Figure 8**. Strong NESs of Nmd3 on Rio2 bypass requirement for Slx9 but not Yrb2 in 40S pre-ribosome export. (**A**) *rio2-nmd3NES* is not synthetic lethal with *mex67Δloop* or *mtr2Δloop116-137*. Strains were spotted in 10-fold serial dilutions on 5-FOA (SD) plates and grown at 30°C for 2–4 days (**B**) The Nmd3-NES (amino acids 440–518) fused to Rio2^ΔNES bypasses the requirement of the Rio2-NES in export complex formation in vitro. GST-Rio2^Nmd3NES was immobilized on GSH-Sepharose and complex formation was carried out and analyzed as in *Figure 2C*. L = input. (**C**) *rio2-nmd3NES* rescues the impaired pre40S export of *slx9Δ* cells. Localization of uS5-GFP in the indicated strains was monitored by fluorescence microscopy. Scale bar = 5 μm. (**D**) The rescue of impaired pre40S ribosome export by *rio2-nmd3NES* is specific for *slx9Δ*. *yrb2Δ* cells transformed with the indicated plasmids was monitored by fluorescence microscopy for the localization of uS5-GFP. Scale bar = 5 μm.

(*Bax et al., 2006*). In support to this early assembly role, *slx9Δ* interacts genetically with mutations in *RRP5*, which encodes an assembly factor required for pre-rRNA processing. However, low-copy overexpression of the transport receptor *MEX67-MTR2*, which is involved in 40S pre-ribosome export, suppresses the growth defect and pre-40S export defect of *slx9Δ* cells. Further, *SLX9* genetically interacts with several factors directly involved in 40S pre-ribosome export (*Faza et al., 2012*). These functional studies indicate a role for Slx9 in the nuclear export of the 40S pre-ribosomal cargo (*Faza et al., 2012*). Thus, the pre-rRNA processing defects observed in the *slx9Δ* mutant could be a consequence of a primary defect in the nuclear export of 40S pre-ribosomes.

Surprisingly, although Slx9 did not interact with Crm1 in vitro, it interacts specifically with the GTP-bound form of Ran. This interaction depends on the basic patch of RanGTP. Removal of the RanGTP acidic C-terminal tail strengthened Slx9:RanGTP interactions, suggesting that these proteins interact in a manner distinct from previously identified RanGTP-binding proteins, such as Yrb1 (*Maurer et al., 2001*; *Nilsson et al., 2001*). Sequence analyses did not reveal any apparent homology between Slx9 and known Ran-binding proteins. Based on all these data, we propose that Slx9 is a new type of RanGTP-binding protein.

## Slx9 mediates non-canonical stepwise assembly of a Crm1-export complex

To assemble a Crm1-export complex, Crm1 requires binding to both RanGTP and a NES-containing cargo in a cooperative manner (*Dong et al., 2009*; *Güttler and Görlich, 2011*; *Monecke et al., 2013*).

This canonical assembly pathway requires Crm1 to switch from a relaxed, low affinity 'cytoplasmic' conformation to a strained, high-affinity 'nuclear' conformation. In the nuclear conformation, RanGTP is enclosed within the toroid-like fold of Crm1, away from the cargo-binding site. Structural analyses of Crm1-complexes suggest that RanGTP promotes NES-binding solely by stabilizing the strained nuclear conformation of Crm1 (*Güttler and Görlich, 2011*; *Monecke et al., 2013*).

In contrast, Slx9 mediates a non-canonical stepwise assembly of a Crm1-export complex. First, Slx9 binds RanGTP and the NES-containing 40S pre-ribosome adaptor Rio2. Interactions between Slx9, Rio2, and RanGTP do not rely on cooperative binding, since the pairwise complexes between these proteins were stable. However, all three proteins were required for efficient Crm1 loading to the Rio2-NES. This recruitment also does not rely on cooperative binding between Rio2, Crm1, and RanGTP, since the Slx9:Rio2:RanGTP complex was stable in the absence of Crm1.

Mutational and functional studies identified a point mutation, L108P, which impaired the ability of Slx9 to bind RanGTP in vitro. Although this mutant protein was stably recruited to pre-ribosomal particles in vivo and to Rio2 in vitro, it was unable to assemble a Crm1-export complex. Both *slx9Δ* and *slx9-1* cells displayed impaired 40S pre-ribosome export and synthetic phenotypes with factors involved in 40S pre-ribosome export. These data suggest that the 40S pre-ribosome nuclear export defect observed in *slx9Δ* and *slx9-1* cells is due to their failure to efficiently assemble Crm1-export complexes on 40S pre-ribosomal particles. A stepwise formation of a Crm1-export complex could allow currently uncharacterized quality control surveillance steps to monitor pre-ribosome assembly and ensure that only correctly assembled particles are chosen for export. In this respect, Slx9-dependent ribosome export would mirror the assembly process of the 40S pre-ribosomal cargo itself.

Crm1-mediated nuclear export is modulated by RanGTP-binding proteins, which promote specific steps along the export pathway. For example, RanBP3/Yrb2 increases the rate of cooperative Crm1-complex assembly in the nucleus (*Koyama et al., 2014*). Yrb2/RanBP3 also modulates Crm1 substrate recognition, promoting interactions with certain NESs and preventing the strong Crm1-Snurportin1 interaction (*Englmeier et al., 2001*; *Langer et al., 2011*). At the other end of the export cycle, RanBP1/Yrb1 and RanBP2 interact strongly with RanGTP in the cytoplasm to stimulate Crm1-export complex disassembly, thereby indirectly contributing to export efficiency (*Bischoff and Görlich, 1997*; *Kehlenbach et al., 1999*; *Maurer et al., 2001*). Notably, despite the fact that these RanGTP-binding proteins act at different stages to stimulate export, they typically influence cooperative interaction between cargo, Crm1, and RanGTP.

Slx9 utilized a distinct mechanism from that proposed for Yrb2/RanBP3 for Crm1-export complex assembly (*Englmeier et al., 2001*; *Lindsay et al., 2001*; *Koyama et al., 2014*). Yrb2/RanBP3 increases the affinity of Crm1 for RanGTP by stabilizing a conformation of Crm1 that promotes canonical assembly of a Crm1-export complex (*Langer et al., 2011*; *Koyama et al., 2014*). In contrast, Slx9 does not interact with RanGTP and Crm1 to form a complex. Instead, it interacts with RanGTP and the NES-containing adaptor Rio2. Finally, biochemical and structural studies show that Yrb2 competes with NES-containing cargo for binding to Crm1 and suggest that NES-binding causes partial dissociation of Yrb2 from the complex (*Koyama et al., 2014*). In contrast, Slx9 remains stably bound to the GST-Rio2:Crm1:RanGTP complex, suggesting that it does not compete with the Rio2-NES for its interaction. Despite these clear differences, both Slx9 and RanBP3/Yrb2 work to overcome the low binding affinity between NESs and Crm1, representing distinct solutions to the paradox of low affinity interactions driving fast and efficient Crm1-mediated cargo export. Notably, *SLX9* strongly genetically interacted with *YRB2*, suggesting that both mechanisms are employed to ensure rapid 40S pre-ribosome export.

Does Slx9 target other NES-containing cargos? Although *slx9Δ* showed a strong 40S pre-ribosome export defect, *rio2ΔNES* grew indistinguishably from WT cells. Notably, *rio2ΔNESyrb2Δ* cells grew slowly (*Figure 7—figure supplement 2B*) but are not synthetically lethal as the *slx9Δyrb2Δ* strain. These genetic interactions argue that Rio2 is not the sole target of Slx9. Genetic approaches will uncover additional NES-containing adaptors that employ Slx9 to prepare the 40S pre-ribosome for nuclear export.

## A family of NES-conformer stabilizers?

NESs contain variability in the spacing between key hydrophobic residues, yet are recognized by the same rigid hydrophobic pockets on Crm1 (*Güttler et al., 2010*). The structure of RanGTP-Crm1 bound to prototypic NESs showed that the backbone of NESs adopts different conformations, which

permits the efficient insertion of key NES residues (*Güttler et al., 2010*). In support of the proposed NES-conformer selection model, a destabilized NES of Rio2$^{3G}$ was unable to bind Crm1 in the presence of RanGTP in vitro. However, Crm1 recruitment to Rio2$^{3G}$ was restored in the presence of Slx9, suggesting a stabilizing function. This stabilization provides a mechanism to overcome the weak binding of NESs to Crm1, driving Crm1-complex assembly and guaranteeing efficient cargo export into the cytoplasm. One prediction of the NES-conformer selection model would be that NESs with favorable conformations exhibit improved affinity to Crm1. Slx9 may belong to a family of yet-unidentified RanGTP-binding proteins that induce and stabilize NES-conformers upon binding export cargos. Thus, these RanGTP-binding proteins would allow greater NES variability and potentially regulate the efficiency by which specific NESs are recognized by Crm1.

## Materials and methods

### Yeast strains and plasmids

Yeast strains used in this study are listed in *Supplementary file 1*. Genomic disruptions, insertion of C-terminal tags, and promotor switches at genomic loci were performed as previously described (*Longtine et al., 1998*; *Puig et al., 2001*; *Janke et al., 2004*). Preparation of media, yeast transformations, and genetic manipulations were performed accordingly to established procedures. Genetic analyses were performed as previously described (*Faza et al., 2012*).

Plasmids used in this study are listed in *Supplementary file 2*. All recombinant DNA techniques were performed accordingly to established procedures using *Escherichia coli* XL1 blue cells for cloning and plasmid propagation.

*rio2ΔNES* was created by deletion of the DNA sequence encoding the last 12 amino acids of Rio2. *ltv1ΔNES* was created by removing the last 13 amino acids. The *rio2-nmd3NES* was created by fusion of the C-terminal region of *NMD3* encoding amino acids 440–518 to *rio2ΔNES*. *rio2-nmd3NESΔ1* was created by fusing *NMD3* lacking amino acids 440–487. The C-terminal DNA-fragment of *SSB1* (*SSB1C*) encodes the amino acids 524–613 (*Maurer et al., 2001*).

Point mutations in *SLX9*, *RIO2*, or *GSP1* were generated using the QuikChange site-directed mutagenesis kit (Agilent Technologies, Switzerland). All cloned DNA fragments generated by PCR amplification and mutagenized plasmids were verified by sequencing.

### Biochemical analysis

Tandem affinity purifications (TAPs) of pre-ribosomal particles were carried out as previously described (*Faza et al., 2012*; *Altvater et al., 2014*). Calmodulin eluates were separated on NuPAGE 4–12% Bis-Tris gradient gels (Invitrogen, Zug, Switzerland). Separated proteins were visualized by either silver staining or Western analysis using indicated antibodies.

Whole cell lysates were prepared by a modified post-alkaline extraction protocol as previously described (*Kemmler et al., 2009*). Extracted proteins were separated by SDS-PAGE. Proteins were visualized by Western blotting using the indicated antibodies.

Western analyses were performed as previously described (*Kemmler et al., 2009*). The following primary antibodies were used: α-Slx9 (1:3000; *Faza et al., 2012*), α-Arc1 (1:4000; E Hurt, University of Heidelberg, Heidelberg, Germany), α-Xpo1 (Crm1) (1:3000; this study), α-His (1:2000; Sigma–Aldrich, USA), α-uS7 (yeast Rps5; 1:4000; Proteintech Group Inc., Chicago, IL, USA), α-TAP (CBP) (1:4000; Thermo Scientific, Rockford, IL, USA), α-Ltv1 (1:5000; K Karbstein, Scripps Research Institute, Jupiter, FL, USA), α-Rio2 (1:1000; Proteintech Group Inc.), α-Tsr1 (1:5000; K Karbstein, Scripps Research Institute), and α-Gsp1 (Ran) (1:3000; this study). For detection, HRP-conjugated α-rabbit (1:2000–1:4000; Sigma–Aldrich) or α-mouse secondary antibodies (1:2000–1:4000; Sigma–Aldrich) were applied. Signals were visualized using the Immun-Star HRP chemiluminescence kit (Bio-Rad Laboratories, Hercules, CA, USA) and captured by Fuji Super RX X-ray films (Fujifilm, Japan).

### Recombinant protein expression and in vitro binding studies

Recombinant Slx9 and Slx9-1, Rio2, Rio2 variants, and Xpo1 (yeast Crm1) were expressed in *E. coli* BL21 upon IPTG induction (final concentration 0.3 mM). His$_6$-tagged proteins were affinity purified in purification buffer (50 mM Tris-HCl, 200 mM NaCl, 10 mM Imidazole, 1 mM β-mercaptoethanol, pH 8), using Ni-NTA agarose (GE Healthcare, Uppsala, Sweden). The GB1-His-domains of these

proteins were removed by TEV cleavage. All proteins were stored in PBS-KMT (150 mM NaCl, 25 mM sodium phosphate, 3 mM KCl, 1 mM $MgCl_2$, 0.1% Tween, pH 7.3) after buffer exchange. GST-fusion proteins were purified in PBSKMT using GSH-Sepharose (GE Healthcare). $His_6$-Gsp1 (Ran) WT and mutants were expressed and purified as previously described (*Solsbacher et al., 1998*; *Maurer et al., 2001*).

For in vitro binding studies, recombinant GST-tagged proteins were immobilized on GSH-Sepharose (GE Healthcare) in PBSKMT and incubated with the indicated proteins for 1 hr at 4°C. Binding studies between GST-tagged Slx9, Yrb1, or Kap123 and Ran variants were performed as described before (*Solsbacher et al., 1998*). To prevent nonspecific interactions, all binding assays were carried out in the presence of competing *E. coli* lysates.

Trimeric export complex formation between GST-tagged Slx9, Rio2, or Ltv1 variants and $His_6$-$Ran^{QL}$GTP and/or Crm1-$His_6$ was adapted and modified from *Rothenbusch et al. (2012)* and *Solsbacher et al. (1998)*. Immobilized GST-fusion proteins were incubated with buffer alone or buffer containing 2 µM $His_6$-$Ran^{QL}$GTP or 50 nM Crm1-$His_6$ or 2 µM $His_6$-$Ran^{QL}$GTP and 50 nM Crm1-$His_6$.

The interactions between GST-fusions of Slx9, Ntf2, and Yrb1 and RanGTP or RanGDP were analyzed as follows: first, $His_6$-Ran was incubated with GDP or GTP (50× molar excess of the protein concentration) in the presence of 6 mM EDTA in $KP_i$ buffer (24.8 mM $KH_2PO_4$, 25 mM, $K_2HPO_4$, 20 mM KCl, 5 mM $MgCl_2$, 2 mM Imidazole, 5 mM β-mercaptoethanol, pH 6.8) on ice for 40 min. This incubation step was terminated by the addition of 0.3 mM $MgCl_2$. In the second step, 2 µM Ran (GTP or GTP) was incubated with immobilized GST-fusion proteins for 1 hr at 4°C.

To form a Rio2:Slx9:Ran:Crm1 complex, GST-Rio2 variants were immobilized on GSH-Sepharose and incubated with buffer alone or 2 µM purified Slx9 variants for 1 hr at 4°C. Then, samples were incubated with 0.2 µM purified $His_6$-$Ran^{QL}$GTP for 1 hr at 4°C. In the last step, samples were incubated with buffer alone or 50 nM purified Crm1-$His_6$. After each incubation step, unbound proteins were removed by three times washing with PBSKMT.

All bound proteins were eluted in LDS-sample buffer (Invitrogen) and separated by SDS-PAGE. Separated proteins were visualized by Coomassie staining or by Western analysis using antibodies against Slx9, Crm1, or Ran. 1/3 of bound proteins and 1/3–3× of input were analyzed on a Coomassie gel. 1/6 of bound proteins and 1/24 (Slx9) or 1/50 (Ran, Crm1, Nmd3, and Rio2) of the input was used for Western analysis.

### Fluorescence microscopy and heterokaryon assay

Cells were visualized using a DM6000B microscope (Leica, Germany) equipped with a HCX PL Fluotar 63×/1.25 NA oil immersion objective (Leica). Images were acquired with a fitted digital camera (ORCA-ER; Hamamatsu Photonics, Japan) and Openlab software (Perkin–Elmer, USA).

Localization of pre-40S subunits was monitored employing the uS5-GFP reporter construct as previously described (*Faza et al., 2012*; *Altvater et al., 2014*). Co-localization of Slx9-GFP and Slx9-1-GFP with Gar1-mCherry was done as previously described (*Faza et al., 2012*).

The heterokaryon assay was adapted and modified from (*Belaya et al., 2006*; *Altvater et al., 2012*). Briefly, equal amounts of cells expressing Enp1-GFP, Gar1-GFP, or Slx9-GFP were mated with *kar1-1* cells expressing Nup82-mCherry and concentrated onto 0.45-µM nitrocellulose filter. Mixtures were placed on YPD plates containing 50 µM cycloheximide. After 1 hr incubation at 30°C, cells were analyzed by fluorescence microscopy.

### FISH

Localization of 20S pre-rRNA in the different strains was analyzed using a Cy3-labeled oligonucleotide probe (5′-Cy3-ATG CTC TTG CCA AAA CAA AAA AAT CCA TTT TCA AAA TTA TTA AAT TTC TT-3′) that is complementary to the 5′ portion of ITS1 as described (*Faza et al., 2012*; *Altvater et al., 2014*).

## Acknowledgements

We are grateful to G Schlenstedt, E Hurt, M Peter, Y Barral, K Karbstein, for generously sharing plasmids, strains, and antibodies. J Pfannstiel for mass spectrometry, all members of the Panse laboratory and in particular C Weirich for enthusiastic discussions.

# Additional information

## Funding

| Funder | Grant reference | Author |
|---|---|---|
| Schweizerische Nationalfonds zur Förderung der Wissenschaftlichen Forschung | | Vikram Govind Panse |
| European Research Council (ERC) | Starting Grant EURIBIO0260676 | Vikram Govind Panse |

The funders had no role in study design, data collection and interpretation, or the decision to submit the work for publication.

## Author contributions

UF, NS, SS, Conception and design, Acquisition of data, Analysis and interpretation of data, Drafting or revising the article, Contributed unpublished essential data or reagents; MA, Acquisition of data, Analysis and interpretation of data, Drafting or revising the article, Contributed unpublished essential data or reagents; YC, Acquisition of data, Analysis and interpretation of data, Contributed unpublished essential data or reagents; MBF, Conception and design, Acquisition of data, Analysis and interpretation of data, Contributed unpublished essential data or reagents; VGP, Conception and design, Drafting or revising the article

# Additional files

## Supplementary files

• Supplementary file 1. Yeast strains used in this study.

• Supplementary file 2. Plasmids used in this study.

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
