## [Decision Letter]

Thank you for sending your work entitled “A non-canonical mechanism for Crm1-mediated nuclear export of 40S pre-ribosomal cargo” for consideration at *eLife*. Your article has been favorably evaluated by James Manley (Senior editor) and three reviewers, one of whom is a member of our Board of Reviewing Editors.

The Reviewing editor and the other reviewers discussed their comments before we reached this decision, and the Reviewing editor has assembled the following comments to help you prepare a revised submission.

This manuscript describes the largely uncharacterized protein Slx9 as a cofactor for Crm1-mediated nuclear export of pre-40S subunits. Whereas much is known about export of the large ribosomal subunit, export of the small subunit has remained enigmatic. It is well established that it depends on Crm1 but essential Crm1 ligands have not been identified. This has led to the suggestion that export factors for the small subunit are redundant. In this work, the authors identify Slx9 as a factor that interacts with Ran-GTP and the NES-containing pre-40S factor Rio2 to recruit Crm1. The data are clearly presented and, for the most part, support the general claims of the authors. However, there are several inconsistencies in the data and interpretation that warrant clarification. While many of these can probably be handled by additional explanations, two points are likely to require new experiments. First, the conclusion of a ternary complex of Slx9-RanGTP-Rio2 is not strongly supported. Second, quantitative interaction analysis rather than the more qualitative GST-pulldown is warranted for certain conclusions. Finally, the text could benefit from a more balanced tone that limits overstatements and overinterpretation of some of the data. Some suggestions are provided in the comments below.

1) In the subsection “Recruitment of Crm1 to Rio2:RanGTP is stimulated by Slx9”, the statement “…a Rio2:Slx9 complex was able to recruit RanGTP, suggesting that Slx9 employs distinct surfaces to bind these two proteins” cannot be supported strongly based on the data. The authors used immobilized Rio2 to pull down either RanGTP directly, or Slx9 followed by RanGTP. The fact they recover RanGTP in the latter could be due to its direct interaction with immobilized Rio2, either because Rio2 was not completely saturated with Slx9, or because RanGTP simply competed off some bound Slx9. It is therefore unclear how this result permits conclusion of a ternary complex. Indeed, in a situation where all pairwise combinations can interact, such GST-pulldowns will not be convincing and other methods are warranted (e.g., gel filtration, gel shift assays, or sequential affinity steps against two of the proteins recovering the third). Alternatively, a well-controlled and quantitative titration of Slx9 to show that it does not inhibit the binding of Ran-GTP could also argue against mutually exclusive binding.

2) Subsection “Recruitment of Crm1 to Rio2:RanGTP is stimulated by Slx9”: from data in Figure 5 the authors claim that Slx9 promotes the formation of a complex containing Rio2, Ran-GTP and Crm1. However, Figure 4 shows that a trimeric complex containing only Rio2-Ran-GTP and Crm1 assembles efficiently in the absence of Slx9. In fact, in that complex, Crm1 appears to enhance Ran binding, suggesting cooperativity of binding. The same components appear to be present in Figure 5 lane 4 but now we do not see Crm1 binding. What accounts for this difference? This is an important point as it underpins the conclusion that Slx9 acts in a non-canonical fashion.

3) The defective interaction between Slx9-1 and RanGTP should be analyzed biophysically to directly compare it to Slx9 (e.g., measurement of Kd values and ideally, on and off rates). The GST pulldown is rather qualitative and the modest difference in recovery observed in Figure 2 (which is not entirely convincing) cannot easily explain why this mutation essentially phenocopies a Slx9∆ strain. As this mutant is used for several key points in the paper, its characterization would appear to be important.

4) The data presented that FAM207A is indeed the human Slx9 homolog as stated (Figure 2–figure supplement 2) is not entirely convincing. The distribution and evolutionary conservation of the Slx9 proteins has not been reported to date. Indeed, it was previously suggested that Slx9 homologs are limited to yeast species (Bax et al RNA 2006). After one referee performed some limited multiple sequence alignments between putative vertebrate FAM207A homologs and the *pombe* and *cerevisiae* Slx9 proteins, he/she did not find the level of amino acid conservation convincing. Particularly in view of the previous report (Bax et al), the authors should show evidence for the phylogenetic distribution and amino acid conservation of the Slx9 proteins beyond yeast. Related to this, the authors show that FAM207A binds Ran^QL^GTP in vitro. However, this observation in isolation is not convincing since cofractionation with ribosomes or requirement for 40S subunit export are not analyzed. The authors should either be more cautious about the claim that FAM207A is the functional human Slx9 homolog or more robustly support this conclusion.

5) In Figure 2, the authors must rule out that newly synthesized protein does not account for the apparent shuttling of Slx9. Notably, the signal for Slx9 in the recipient nucleus seems less than that for the positive control of Enp1.

Minor comments:

1) In their previous paper (17) the Panse group show that at 37 °C, slx9Δ yeast cells grow almost like wild type with restoration of 40S subunit export. How do the authors square this apparent paradox with the findings reported in the current manuscript? Do cells carrying the slx9-1 allele also show restoration of 40S export at 37 °C? Are the proposed “weak” binding events such as the Rio2-Crm1 interaction temperature-dependent?

2) How does the proposed role of Slx9 in Crm1-mediated nuclear export relate to the previously reported role of Slx9 in ITS1 pre-rRNA processing (Bax et al RNA 2006)? This previous work has not been referenced and should be discussed in the text.

3) I am a little confused by the data in Figure 2 and Figure 5. In Figure 2, the authors claim that the interaction between GST-Slx9-1 and His6-Ran^QL^GTP is reduced compared with wild type GST-Slx9. However, to my eye, there seems to be less GST-Slx9-1 compared with GST-Slx9. Indeed, in contrast with Figure 2, Slx9 and Slx9-1 seem to bind similarly to Ran^QL^GTP in Figure 5. Is this apparent difference in Slx9-1-Ran^QL^GTP binding due to the presence of Rio2 in Figure 5?

4) There is a lot of discussion in the text about “weak” (Rio2) and “strong” (Nmd3) NES sequences in Crm1-mediated export. Can the authors define this more quantitatively.

5) In Figure 1 the authors report that slx9-1 localizes to the nucleolus, like the wild-type protein but they claim that it blocks export, evidenced by accumulation of pre-40S markers in the nucleoplasm. If Slx9-1 loads onto particles but prevents their recruitment, its localization should be coincident with the pre-40S markers. How do the authors explain this discrepancy?

6) The authors identified *slx9-1* as a loss of function mutant and showed that it had reduced interaction with Ran-GTP. In light of that, it is surprising that it appears to bind more tightly to Rio2 and recruit higher levels of Ran-GTP. How can this be explained?

7) The authors should explain what the biophysical property of GGG is and how they imagine this will affect the presentation of the NES. What do they mean that the GGG mutant “destabilizes” the NES?

8) Figure 5 is intended to show that Slx9-1 interacts more strongly than Slx9 (i.e., lane 6 versus 2, respectively). While this is apparent on the immunoblot, inspection of the stained gel reveals little if any difference. Similarly, the input lanes show Slx9 and Slx9-1 at seemingly equal levels on the stained gel, yet give clearly different signals by immunoblot. This raises the somewhat worrisome possibility that differences see in Figure 1 (and elsewhere) are a blotting artifact, which is remarkably common when comparing proteins that migrate differently or are mutated. Such potential discrepancies are worth clarifying.

9) In the subsection “Slx9 acts as a scaffold to load Crm1 onto the Rio2-NES”, the authors state that the RanGTP binding site of Slx9 plays a role in recruiting Crm1, but the basis for this claim is not clear given that the authors do not know what/where the RanGTP binding site is on Slx9.

10) I am not sure what the basis of the ‘stepwise assembly’ claim is. I realize they have often biochemically performed the reactions in a certain order, but given the various pairwise interactions that are shown in the paper, I cannot see how an order is forthcoming (beyond Crm1 arriving last). The authors should either provide a clear rationale for a specific assembly order, or remain more agnostic on the issue.

11) The authors make a number of statements regarding kinetic parameters such as cooperativity, but I fail to see how many of these claims can be supported on the basis of relatively qualitative GST-pulldown assays.

12) Words such as ‘unexpectedly’ are not all that useful in the Abstract since it is unclear what was unexpected or why, without further explanation. In general, I struggled with the repeated claims of novelty and non-canonical mechanisms when the concept appeared to this reader as a variation on that seen with RanBP3/Yrb2. A less exaggerated tone seems preferable, although I realise different authors will have different writing styles.

---

## [Author Response]

*This manuscript describes the largely uncharacterized protein Slx9 as a cofactor for Crm1-mediated nuclear export of pre-40S subunits. Whereas much is known about export of the large ribosomal subunit, export of the small subunit has remained enigmatic. It is well established that it depends on Crm1 but essential Crm1 ligands have not been identified. This has led to the suggestion that export factors for the small subunit are redundant. In this work, the authors identify Slx9 as a factor that interacts with Ran-GTP and the NES-containing pre-40S factor Rio2 to recruit Crm1. The data are clearly presented and, for the most part, support the general claims of the authors. However, there are several inconsistencies in the data and interpretation that warrant clarification. While many of these can probably be handled by additional explanations, two points are likely to require new experiments. First, the conclusion of a ternary complex of Slx9-RanGTP-Rio2 is not strongly supported. Second, quantitative interaction analysis rather than the more qualitative GST-pulldown is warranted for certain conclusions. Finally, the text could benefit from a more balanced tone that limits overstatements and overinterpretation of some of the data. Some suggestions are provided in the comments below*.

We thank the reviewers for their thorough and fair comments. We have tried to address these three points.

*1) In the subsection “Recruitment of Crm1 to Rio2:RanGTP is stimulated by Slx9”, the statement “…a Rio2:Slx9 complex was able to recruit RanGTP, suggesting that Slx9 employs distinct surfaces to bind these two proteins” cannot be supported strongly based on the data. The authors used immobilized Rio2 to pull down either RanGTP directly, or Slx9 followed by RanGTP. The fact they recover RanGTP in the latter could be due to its direct interaction with immobilized Rio2, either because Rio2 was not completely saturated with Slx9, or because RanGTP simply competed off some bound Slx9. It is therefore unclear how this result permits conclusion of a ternary complex. Indeed, in a situation where all pairwise combinations can interact, such GST-pulldowns will not be convincing and other methods are warranted (e.g., gel filtration, gel shift assays, or sequential affinity steps against two of the proteins recovering the third). Alternatively, a well-controlled and quantitative titration of Slx9 to show that it does not inhibit the binding of Ran-GTP could also argue against mutually exclusive binding*.

We agree with the reviewers, and have done titrations as suggested. Specifically, we added increasing amounts of RanGTP to a preformed saturated GST-Rio2:Slx9 (or GST-Slx9:Rio2) complex. RanGTP is efficiently recruited to both GST-Rio2:Slx9 and GST-Sxl9:Rio2 complexes. Further, RanGTP binding did not displace Slx9 from the GST-Rio2:Slx9 complex (or Rio2 from GST-Slx9:Rio2). These data are now included in Figure 5.

In addition to these experiments, we believe that two different observations strongly support the notion for distinct binding surfaces:

1. Although Rio2 can bind RanGTP or Slx9 on its own, neither the GST-Rio2:RanGTP complex nor the GST-Rio2:Slx9 complex is competent to recruit Crm1 (Figure 6, lane 4 and Figure 6, lane 4). Only an immobilized GST-Rio2 that is sequentially treated with Slx9 and RanGTP can recruit Crm1 (Figure 6, lane 8).

2. The GST-Rio2^3G^ mutant is unable to cooperatively bind to Crm1 and RanGTP, (Figure 7, lane 4). Only when GST-Rio2^3G^ is sequentially treated with Slx9 and then RanGTP, the resultant GST-Rio2^3G^:Slx9:RanGTP complex is now competent to recruit Crm1 (Figure 7, lane 4).

Together these data support the idea that GST-Rio2:Slx9:RanGTP form a complex.

*2) Subsection “Recruitment of Crm1 to Rio2:RanGTP is stimulated by Slx9”: From data in*
Figure 5
*the authors claim that Slx9 promotes the formation of a complex containing Rio2, Ran-GTP and Crm1. However,*
Figure 4
*shows that a trimeric complex containing only Rio2-Ran-GTP and Crm1 assembles efficiently in the absence of Slx9. In fact, in that complex, Crm1 appears to enhance Ran binding, suggesting cooperativity of binding. The same components appear to be present in*
Figure 5
*lane 4 but now we do not see Crm1 binding. What accounts for this difference? This is an important point as it underpins the conclusion that Slx9 acts in a non-canonical fashion*.

We agree with the reviewers that this is confusing. As the reviewers correctly points out, in Figure 4, lane 4, Crm1 and excess RanGTP were added simultaneously to GST-Rio2. In this experiment, Rio2, RanGTP and Crm1 form a trimeric complex through a cooperative mechanism. In the absence of Crm1, RanGTP shows a weaker binding to Rio2 (Figure 4, lane 2).

In Figure 6, the experimental setup is different (summarized in the flow chart, Figure 6). In a first step, RanGTP is added to GST-Rio2 and then unbound protein is washed away. Again, we observed binding to RanGTP (Figure 6, lane 2 and 3), in agreement with Figure 4. Then, in a second step, Crm1 is incubated with a preformed GST-Rio2:RanGTP complex (Figure 6, lane 4). However, because Crm1 is added in the absence of excess of RanGTP, no Crm1 binding is observed. Thus, presenting Crm1 to a Rio2:RanGTP complex in an ordered fashion does not allow cooperative binding. In this stepwise mechanism, Slx9 aids Crm1 recruitment to the GST-Rio2:Slx9:RanGTP complex by optimally presenting the Rio2-NES and RanGTP.

We have altered the discussion of this data to make this distinction more clear.

*3) The defective interaction between Slx9-1 and RanGTP should be analyzed biophysically to directly compare it to Slx9 (e.g., measurement of Kd values and ideally, on and off rates). The GST pulldown is rather qualitative and the modest difference in recovery observed in*
Figure 2
*(which is not entirely convincing) cannot easily explain why this mutation essentially phenocopies a Slx9∆ strain. As this mutant is used for several key points in the paper, its characterization would appear to be important*.

We attempted two quantitative assays, as described above: ITC and SPR. Unfortunately, these experiments were unsuccessful due to technical issues. Specifically, we were unable to attain the necessary concentrations of Slx9 and RanGTP required for ITC due to aggregation, and Slx9 displayed non-specific interactions with immobilized GST that is required to immobilize RanGTP on the surface of the SPR chip. Instead, we have quantified the binding experiment between Slx9 (or Slx9-1) and RanGTP from four independent experiments (Figure 2). Specifically, we have normalized the RanGTP Western signal to the amount of GST-Slx9 and GST-Slx9-1 in the Coomassie-stained gel above. These quantifications confirm that GST-Slx9-1 is impaired in RanGTP binding compared to Slx9 (approximately 50% decrease). Although this is not a definitive method to measure affinity, we hope that the review agrees that the 50% reduction in binding is modest. We have tried to clarify this in the text, in the subsection headed “Slx9-1 is impaired in binding Ran^QL^GTP”.

We apologize for the confusion regarding the *slx9-1* mutant phenotype. Two different complexes (outlined below) containing Slx9-1 were impaired in recruiting Crm1 in vitro. This defect is far more pronounced than the defect in Slx9-1 binding to RanGTP and, we believe that this is the major determinant of the Slx9-1 phenotype in vivo.

1) Slx9-1 is impaired in recruiting Crm1 to the GST-Rio2:Slx9-1:RanGTP complex (Figure 6, compare lane 5 to lane 9).

2) The Rio2^3G^ mutant is unable to cooperatively bind to Crm1 and RanGTP (Figure 7, lane 4). Only when bound to Slx9 and RanGTP it is now able to efficiently recruit Crm1 (Figure 7, lane 4). Consistent with the role of Slx9 in optimally presenting the Rio2-NES to Crm1, the GST-Rio2^3G^:Slx9-1:RanGTP complex is unable to bind Crm1 (Figure 7, compare lane 1 and 2). This additional experiment further supports the idea that Slx9 promotes Crm1 recruitment to the GST-Rio2^3G^:Slx9:RanGTP complex.

*4) The data presented that FAM207A is indeed the human Slx9 homolog as stated (Figure 2–figure supplement 2) is not entirely convincing. The distribution and evolutionary conservation of the Slx9 proteins has not been reported to date. Indeed, it was previously suggested that Slx9 homologs are limited to yeast species (Bax et al RNA 2006). After one referee performed some limited multiple sequence alignments between putative vertebrate FAM207A homologs and the* pombe *and* cerevisiae *Slx9 proteins, he/she did not find the level of amino acid conservation convincing. Particularly in view of the previous report (Bax et al), the authors should show evidence for the phylogenetic distribution and amino acid conservation of the Slx9 proteins beyond yeast. Related to this, the authors show that FAM207A binds Ran*^*QL*^*GTP in vitro. However, this observation in isolation is not convincing since cofractionation with ribosomes or requirement for 40S subunit export are not analyzed. The authors should either be more cautious about the claim that FAM207A is the functional human Slx9 homolog or more robustly support this conclusion*.

We agree with the reviewers and have removed the data concerning FAM207A from the manuscript. We originally chose FAM207A (also called C21orf70) as the potential human homolog of *S. cerevisiae* Slx9 because it is listed as the human homologue for *S. pombe* Slx9 in PomBase database (http://www.pombase.org/spombe/result/SPAPB21F2.03). In addition, FAM207A interacts with pre-ribosomal particles and is impaired in pre-40S export (Wyler et al., 2011 RNA). However, we have also been unable to align *S. cerevisiae* Slx9 and human FAM207A with any significance, and share the doubt that these two are not true functional homologues.

5) In Figure 2, the authors must rule out that newly synthesized protein does not account for the apparent shuttling of Slx9. Notably, the signal for Slx9 in the recipient nucleus seems less than that for the positive control of Enp1.

We have repeated the heterokaryon assay in presence of 50 µM cycloheximide which inhibits production of newly synthesized Slx9 protein. These data show that Slx9-GFP shuttles between the nuclear and cytoplasmic compartments (Figure 2).

Regarding the difference in recipient nuclear signal, we suspect that there are two pools of Slx9, only one of which can shuttle. *SLX9* interacts genetically with *SGS1*, which encodes a nucleolar DNA helicase that prevents genomic instability (Ooi et al., Nature Genetics 2003). This additional nucleolar role could explain why there is only partial transfer of Slx9-GFP to the recipient nucleus. This additional role could also explain the primarily (though not exclusively) nucleolar localization of Slx9 and Slx9-1.

Minor comments:

*1) In their previous paper (*[17]*) the Panse group show that at 37 °C, slx9Δ yeast cells grow almost like wild type with restoration of 40S subunit export. How do the authors square this apparent paradox with the findings reported in the current manuscript? Do cells carrying the slx9-1 allele also show restoration of 40S export at 37 °C? Are the proposed “weak” binding events such as the Rio2-Crm1 interaction temperature-dependent*?

Like the *slx9∆* mutant, the *slx9-1* allele also grows normally at 37°C and is also not impaired in the nuclear export of 40S pre-ribosomes. Indeed, this “cold-sensitive” phenotype is relatively common among ribosome synthesis factor mutants. We currently do not understand the basis of the temperature dependence, or whether the binding between Rio2 and Crm1 is temperature-dependent in vivo or, for example, whether ribosome export becomes less dependent on Rio2 at high temperatures. For completeness, we have now added the 37°C growth panel to Figure 1.

2) How does the proposed role of Slx9 in Crm1-mediated nuclear export relate to the previously reported role of Slx9 in ITS1 pre-rRNA processing (Bax et al RNA 2006)? This previous work has not been referenced and should be discussed in the text.

We apologize for the oversight and have now included this important reference in the Discussion. However, we believe that our data are compatible with [5] RNA and that the ITS1 pre-rRNA processing defect caused by Slx9 depletion could be a secondary consequence of the primary defect in export, since the growth and the pre-40S export defect of the *slx9∆* mutant can be rescued by low copy overexpression of an additional export factor Mex67-Mtr2 ([17] PLOS Genetics).

*3) I am a little confused by the data in*
Figure 2
*and*
Figure 5*. In*
Figure 2*, the authors claim that the interaction between GST-Slx9-1 and His6-Ran*^*QL*^*GTP is reduced compared with wild type GST-Slx9. However, to my eye, there seems to be less GST-Slx9-1 compared with GST-Slx9. Indeed, in contrast with*
Figure 2*, Slx9 and Slx9-1 seem to bind similarly to Ran*^*QL*^*GTP in*
Figure 5*. Is this apparent difference in Slx9-1-Ran*^*QL*^*GTP binding due to the presence of Rio2 in*
Figure 5?

The reviewers are correct that these data are somewhat somehow confusing. In Figure 2, pull down of RanGTP by GST-Slx9 or GST-Slx9-1 is shown, and Slx9-1 has a modestly reduced interaction compared to Slx9. We have resolved the issue with loading of GST-Slx9 vs. GST-Slx9-1 by quantification of four independent experiments (presented in Figure 2). Specifically, we have normalized the RanGTP signal in the Western blot to the amount of GST-Slx9 and GST-Slx9-1 in the Coomassie-stained gel above. These quantifications confirm that GST-Slx9-1 is impaired in RanGTP binding compared to Slx9 (approximately 50% decrease). Although this is not a definitive method to measure affinity, we hope that the review agrees that the 50% reduction in binding is modest.

In Figure 5, the immobilized protein is GST-Rio2, which has been pre-loaded with either Slx9 or Slx9-1. In this experiment, as pointed out by the reviewers, RanGTP binds equally well to both complexes. However, the complex containing GST-Rio2/Slx9-1/RanGTP is impaired in recruiting Crm1. Therefore, the reviewers are correct in that the difference between these two experiments is the presence of Rio2, which can also bind RanGTP on its own.

*4) There is a lot of discussion in the text about “weak” (Rio2) and “strong” (Nmd3) NES sequences in Crm1-mediated export. Can the authors define this more quantitatively*.

The difference in NES strength is based on the ability of Crm1 to bind the NES sequence (even without RanGTP; [15] EMBO J). This interaction was shown to correlate with sequence similarity to a consensus NES sequence ([15] EMBO J), and has been shown to correlate with affinity for Crm1 in vitro ([43] TIBS; [30] NSMB). We have altered the text both to include this historical categorization as well as acknowledged the prior work, and we apologize for the oversight.

*5) In*
Figure 1
*the authors report that slx9-1 localizes to the nucleolus, like the wild-type protein but they claim that it blocks export, evidenced by accumulation of pre-40S markers in the nucleoplasm. If Slx9-1 loads onto particles but prevents their recruitment, its localization should be coincident with the pre-40S markers. How do the authors explain this discrepancy*?

Slx9 interacts genetically with a nucleolar DNA helicase, Sgs1 that prevents genomic instability (Ooi et al, Nature Genetics 2003). This additional role which would be strictly nucleolar, would be consistent with the primarily (though not exclusively) nucleolar localization of Slx9 and Slx9-1.

*6) The authors identified* slx9-1 *as a loss of function mutant and showed that it had reduced interaction with Ran-GTP. In light of that, it is surprising that it appears to bind more tightly to Rio2 and recruit higher levels of Ran-GTP. How can this be explained*?

The reviewers are correct that, in the context of Rio2, Slx9-1 binds Ran-GTP more efficiently (Figure 6). Our statement that Slx9-1 is impaired in RanGTP binding stems from the data shown in Figure 2, which is a pull down of RanGTP using GST-Slx9 or GST-Slx9-1, which were performed in the absence of Rio2.

We do observe that Slx9-1 binds more strongly to Rio2 than Slx9 (Figure 6, compare lanes 2 and 6). In addition, the GST-Rio2:Slx9-1 complex appears to bind more strongly to RanGTP than the GST-Rio2:Slx9 complex (Figure 6, compare lanes 3 and 7). However, we also observe that Rio2 alone also binds RanGTP (eg. Figure 6, lane 2). Since, in these experiments, there are two potential RanGTP binding sites (one on Slx9 and one on Rio2), it is difficult to conclude anything regarding the affinity between Slx9/Sxl9-1 and RanGTP in these pull downs.

7) The authors should explain what the biophysical property of GGG is and how they imagine this will affect the presentation of the NES. What do they mean that the GGG mutant “destabilizes” the NES?

We apologize for the oversight for not explaining the use of glycine mutants clearly. Because glycine residues lack a side chain, they allow greater conformational flexibility for the polypeptide backbone of these residues (Ramachandran G N, Sasisekharan V, 1968 Adv Protein Chem) as well as to the neighboring NES, thus destabilizing it. Therefore, introduction of glycines are expected to lower the affinity of an NES for Crm1 by locally disturbing the secondary structure that is important to orient and insert the residues of the NES into the hydrophobic pockets of Crm1 ([30] NSMB). A short explanation has now been included in the text.

*8)*
Figure 5
*is intended to show that Slx9-1 interacts more strongly than Slx9 (i.e., lane 6 versus 2, respectively). While this is apparent on the immunoblot, inspection of the stained gel reveals little if any difference. Similarly, the input lanes show Slx9 and Slx9-1 at seemingly equal levels on the stained gel, yet give clearly different signals by immunoblot. This raises the somewhat worrisome possibility that differences see in*
Figure 1
*(and elsewhere) are a blotting artifact, which is remarkably common when comparing proteins that migrate differently or are mutated. Such potential discrepancies are worth clarifying*.

We agree with the reviewers that there is a slightly stronger signal for Slx9-1 in the Western blot. We have analyzed the differences by carefully running equalized samples of the two proteins, and this difference is small, but present (Figure 9). However, whether Slx9-1 binds slightly better to GST-Rio2 in vitro or to the pre-ribosomal particle (Enp1-TAP) in vivo (Figure 1) does not change the major conclusion of our experiments: Slx9-1 is expressed *in vivo* and recruited to the 40S pre-ribosome. Therefore, we have altered the description of the interaction studies to say that Slx9-1 binds at least as well as the wild-type.

Author response image 1.Comparison of Coomassie staining and Western blot signals of Slx9 and Slx9-1 proteins. 0.5 µM Slx9 or 0.5 µM Slx9-1 were separated on SDS-PAGE (top panel) and either stained with Coomassie or detected by Western analysis with Slx9 antibody (lower panel). L = input.**DOI:**
http://dx.doi.org/10.7554/eLife.05745.017

*9) In the subsection “Slx9 acts as a scaffold to load Crm1 onto the Rio2-NES”, the authors state that the RanGTP binding site of Slx9 plays a role in recruiting Crm1, but the basis for this claim is not clear given that the authors do not know what/where the RanGTP binding site is on Slx9*.

We agree with the reviewers that we currently do not know where the RanGTP binding site is on Slx9. However, our statement regarding the impaired binding is based on the data shown in Figure 2 (which shows that Slx9-1 on its own is impaired in binding RanGTP as compared to wild-type Slx9) and the data in Figure 6, which shows that, although the Rio2: Slx9-1 complex can recruit RanGTP, the resulting complex is significantly impaired in recruit Crm1. Since this defect is not due to the interaction between Rio2 and Slx9-1, and because Slx9 does not interact with Crm1 (Figure 2) we propose that the defect must stem from RanGTP binding by Slx9.

*10) I am not sure what the basis of the ‘stepwise assembly’ claim is. I realize they have often biochemically performed the reactions in a certain order, but given the various pairwise interactions that are shown in the paper, I cannot see how an order is forthcoming (beyond Crm1 arriving last). The authors should either provide a clear rationale for a specific assembly order, or remain more agnostic on the issue*.

The stepwise assembly model that we favor is based on the *in vitro* interactions that we have identified and the cellular context in which these interactions must occur. Slx9 at steady state is localized to the nucleolus/nucleoplasm and Rio2 is nucleoplasmic/cytoplasmic (mainly cytoplasmic). Further, based on TAP purifications and Western analyses Slx9 is recruited to the earliest 40S pre-ribosome in the nucleolus. Once the pre-ribosome becomes nuclear Rio2 and RanGTP join, followed by Crm1.

11) The authors make a number of statements regarding kinetic parameters such as cooperativity, but I fail to see how many of these claims can be supported on the basis of relatively qualitative GST-pulldown assays.

Our use of the term “cooperativity” is based on previous studies, which have shown that the binding of RanGTP and the NES containing cargo to Crm1 is cooperative i.e. the affinity for each binding partner is increased in the presence of the other. For example as shown in Figure 2: The NES of SsbC does not bind Crm1 alone, but only in the presence of excess of RanGTP.

In this manuscript, we show that such a cooperative mechanism cannot be behind the recruitment of Crm1 to the Rio2-Slx9-RanGTP complex, since RanGTP and Crm1 are added in separate steps (so the affinity of the first protein can’t be improved by the presence of the second protein). Although we have been unable to perform more quantitative binding assays, our findings support a stepwise mechanism rather than cooperativity, and therefore we believe that these assays have merit.

*12) Words such as ‘unexpectedly’ are not all that useful in the Abstract since it is unclear what was unexpected or why, without further explanation. In general, I struggled with the repeated claims of novelty and non-canonical mechanisms when the concept appeared to this reader as a variation on that seen with RanBP3/Yrb2. A less exaggerated tone seems preferable, although I realise different authors will have different writing styles*.

We have removed “unexpectedly” from the Abstract and altered the tone of the discussion as suggested by the reviewers.

We do not quite agree with the reviewers that the concept we are promoting is a variation to RanBP3/Yrb2 for the following reasons:

A) Yrb2 binds RanGTP and Crm1 in a cooperative manner and then the complex is presented to the NES containing cargo. In our case, Slx9 binds RanGTP and the NES containing cargo, and this complex is able to recruit Crm1 directly.

B) Yrb2 functions by populating the strained Crm1:RanGTP bound conformation that favors NES binding. Here, we present different mutational studies that suggest that within the Rio2:Slx9:RanGTP complex, Slx9 optimally orients the NES and improves its affinity for Crm1. For example, Rio2^3G^ mutant is unable to cooperatively bind Crm1 and RanGTP is able to do so, when in complex with Slx9.